# Photosynthetic Homeostasis Mechanism and Configuration Application of Woody Plants in Green Wall Under Light Gradients of Building Facades with Different Orientations

**DOI:** 10.3390/plants14233570

**Published:** 2025-11-22

**Authors:** Qiang Xing, Dongfan Xu, Hongbing Wang, Jun Qin, Nannan Dong, Bin Zhao, Yonghong Hu

**Affiliations:** 1Urban Horticulture Research and Extension Center, Shanghai Chenshan Botanical Garden, Shanghai 201602, China; xingqiang0731@126.com (Q.X.); qinjun03@126.com (J.Q.); 2School of Life Science, Fudan University, Shanghai 200433, China; dfxu22@m.fudan.edu.cn; 3College of Life Sciences, Shanghai Normal University, Shanghai 200234, China; whb0236@sina.com; 4College of Architecture and Urban Planning, Tongji University, Shanghai 200092, China; dongnannan@tongji.edu.cn

**Keywords:** vertical greenery, light gradients, photosynthetic characteristics, light-adaptability screening

## Abstract

Under the dual pressures of urbanization and climate change, vertical greening plays a crucial role in compensating for limited urban green space and in enhancing both landscape quality and ecological functions. To improve plant selection strategies for vertical greening, this study quantified the photosynthetic adaptability of ten green wall species across light gradients on all four building facades. Over three years of in situ monitoring, combined with analyses of photosynthetic parameter variation (P_max_, LCP, and LSP) and biomass, we evaluated the physiological and ecological response mechanisms of plants with different growth forms at multiple scales (“trait–species–community”). The main findings are as follows: (1) Photosynthetically active radiation intensity had the order of south wall > east wall > west wall > north wall, with differences of 3–5 fold. (2) Cluster analysis of photosynthetic traits revealed a sensitivity hierarchy: monocotyledonous herbs > dicotyledonous herbs and vines > woody species. Woody plants such as *Ligustrum sinense*, *Ligustrum japonicum*, and *Rhododendron* spp. showed significantly lower variability in photosynthetic parameters (45.8–64.5%) than herbaceous species, thereby maintaining strong stability under light gradients. *Muehlenbeckia complexa* (Mc) adapted to intense south-facing light, while *Acorus gramineus* ‘Ogon’ (Ag) preferred the weaker light of north-facing walls. In contrast, *Farfugium japonicum* (Fj), *Carex oshimensis* (Co), *Trachelospermum jasminoides* (Tj), and *Vinca major* (Vm) displayed substantial physiological fluctuations. (3) Based on three years of monitoring data, we developed a quantitative model of light adaptation driven by the coefficient of variation (CV) of photosynthetic parameters. Together with PCA-based clustering, we proposed a “growth form–orientation” synergistic configuration framework. Woody plants with high photosynthetic stability are recommended as the structural backbone for cost-efficient green walls, supplemented by vine/herbaceous species selected according to wall orientation. This study not only provides a scientific basis for accurate plant selection and low-maintenance design of green walls but also provides technical strategies for integrating multifunctional green infrastructure with low-carbon urban development. Furthermore, the proposed approach can be standardized as a Nature-Based Solutions (NbS) pathway for widespread application to building facades in high-density cities worldwide.

## 1. Introduction

Urbanization poses severe ecological challenges, with 56% of the global population residing in cities—a figure projected to reach 75% by 2050 [1]. This expansion drives the conversion of ecological land to meet construction demands, exacerbating urban heat island effects and energy crises [2,3,4]. Consequently, transforming buildings from “energy-consuming units” into “ecological complexes” is imperative. Green walls, as core components of Nature-based Solutions (NbS), offer 3–5 times greater spatial efficiency than green roofs and can synergistically enhance carbon sequestration, biodiversity, and thermal regulation [5,6]. However, global implementation rates remain below 5% [7], primarily due to poor plant adaptability leading to short lifespans, weak ecological performance, and high maintenance costs [8,9,10]. Thus, quantifying how light environments influence plant performance is critical to unlocking their full ecological benefits.

Current research on green wall reveals three major limitations, with the lack of quantitative comparisons between woody and herbaceous species representing a pivotal gap. Technical Bias: Studies often prioritize water and fertilizer inputs to enhance morphological traits (roots, stems, and leaves) while neglecting photosynthesis—the foundation of ecosystem services [11,12,13,14]. Most research on photosynthesis focuses on improving plant productivity per unit space. In urban green walls, survival and overall persistence are far more important than the production of assimilates. Photosynthetic efficiency is inherently heterogeneous across urban environments, influenced by light, temperature, water, and nutrients [15,16,17]. For instance, the same species exhibits significant photosynthetic variation across orientations (e.g., differing PAR responses), and distinct species accumulate assimilates disparately under identical PAR exposure [18]. However, quantitative data comparing woody and herbaceous species’ photosynthetic parameters (P_max_, LCP, and LSP variability) are absent, hindering mechanistic understanding. Woody Plant Knowledge Void: Practice is dominated by herbaceous and climbing species, with scant understanding of woody species’ applicability. Woody plants potentially enhance structural diversity through canopy stratification, provide year-round coverage as physical barriers against environmental stresses (wind, rain, snow), and ensure long-term stability of services like cooling and rainwater use. However, comparative metrics—such as biomass allocation, root-to-shoot ratios, or photosynthetic stability under light gradients—between woody and herbaceous species remain unquantified, limiting evidence-based species selection. Absence of Quantitative Standards: Configurations rely on empirical ecological classifications (sun-loving/shade-tolerant) rather than species-level light adaptation data [19,20]. This impedes robust evaluation criteria, as key contrasts—such as lower photosynthetic parameter variability in woody species versus herbs, or differential energy benefits—are not measured. Without quantitative thresholds, practical applications remain speculative.

Drawing on the differences in photosynthetically active radiation (PAR) across orientations of the largest green wall in Shanghai, this study examined the light-response mechanisms of ten plant species with distinct growth forms under four-sided light gradients. Our objective was to quantify leaf-level photosynthetic parameters and individual adaptability in order to optimize the ecological functions of green wall plant communities. We hypothesized that (1) variation in light conditions among wall orientations would significantly affect photosynthetic parameters and biomass traits; (2) plant growth forms would differ in their responsiveness to light, with herbaceous species exhibiting strong tillering ability and rapid recovery that allow faster adjustment to fluctuations in wall environments, vines showing intermediate adaptability, and woody plants displaying the greatest stability; and (3) woody species would tolerate a broader PAR range than herbs and vines, maintaining more consistent performance in heterogeneous light environments and thus serving as model species for light adaptation.

## 2. Results

### 2.1. Patterns of Microclimate Differentiation in Facade Light Gradient Environments Across Building Orientations

Figure 1 illustrates the patterns of key environmental factors affecting photosynthesis in green wall plants—photosynthetically active radiation (PAR), air temperature, and CO_2_ concentration. Air temperature ranged from 31.8 to 37.8 °C, with no significant differences among wall orientations (Figure 1I). CO_2_ concentration exhibited a decreasing-then-increasing trend, and values across different orientations showed no significant differences at the measured times (Figure 1II).

For PAR, all wall orientations generally displayed an initial increase followed by a decrease throughout the day. However, significant differences in PAR levels were observed at different times for each orientation. The timing of peak PAR varied: the north wall peaked at 14:00; the east wall at approximately 11:00; and the south and west walls both around 12:00 (Figure 1III). The duration and timing of high PAR (above 1500 μmol·m^−2^·s^−1^) also differed among orientations: the east wall experienced ~3 h from 9:00 to 11:00; the south wall ~5 h from 9:00 to 13:00; the west wall ~2 h from 12:00 to 13:00; and the north wall ~2 h from 14:00 to 15:00.

Overall, the average PAR levels across orientations decreased in the following order: south wall > east wall > west wall > north wall. The south wall differed significantly from the north wall, representing a clear contrast between high-light and low-light environments. The north wall showed drastic and unstable fluctuations from low to high to low light. The east and west walls exhibited similar light conditions, both significantly different from the south and north walls (Figure 2).

### 2.2. Light Response Characteristics of Green Wall Plants in Buildings with Different Orientations

Light response curves were measured for 10 plant species, and four primary photosynthetic parameters were derived from these curves to assess adaptive differences under varying light environments. South-facing and north-facing walls represented strong and weak light conditions, respectively, while east-facing and west-facing walls, with less pronounced differences, were used for further refinement.

As shown in Figure 3, the photosynthetic rate of all 10 species generally increased initially and then stabilized with rising photosynthetically active radiation (PAR), although species-specific differences were observed. For Ag, Fj, and Tj, the plateau phase on the north wall was higher than on the south wall, indicating adaptation to shaded, north-facing environments. For Ls, Lj, Rp, and Ri, the plateau phases were similar between north- and south-facing walls, reflecting stable performance under contrasting light conditions. In contrast, Co, Mc, and Vm exhibited lower plateau values on the north wall compared to the south wall, suggesting a preference for sun-exposed, south-facing environments.

Further comparison under similar total light conditions on east- and west-facing walls revealed additional species-specific adaptations. Stable growth segments for Ag, Fj, and Ls were higher on the west wall than on the east wall, indicating a tendency toward shade tolerance. Conversely, stable growth segments for Mc, Rp, and Vm were lower on the west wall than on the east wall, suggesting sun tolerance and adaptation to east-facing environments. Stable growth periods for Co, Lj, Ri, and Tj were largely consistent between the east and west walls.

### 2.3. Photosynthetic Parameters and Biomass Characteristics of 10 Green Wall Plants Under Different Orientations

By fitting the light response curves of 10 plant species under different wall orientations, five photosynthetic parameters were obtained: apparent quantum efficiency (AQE), maximum photosynthetic rate (P_max_), light compensation point (LCP), light saturation point (LSP), and dark respiration rate (R_d_). These parameters were used to assess and classify the relative photosynthetic adaptation of each species.

As shown in Figure 4I, analysis of AQE indicated that Fj and Ri exhibited higher values on the north wall compared to the south wall, whereas Mc, Tj, and Vm had significantly higher values on the south wall. For the remaining species, no significant differences were observed between the north and south walls. Regarding east–west comparisons, only Ls showed significantly higher AQE on the west wall than the east wall, while Co and Mc exhibited lower values on the west wall.

Figure 4II illustrates P_max_ patterns. Ag and Tj had higher values on the north wall, whereas Fj, Mc, and Vm showed higher values on the south wall. No significant differences were observed for the other five species. In the east–west comparison, Ag, Fj, and Ls exhibited significantly higher P_max_ on the west wall, while Mc and Rp were higher on the east wall; the remaining five species showed no significant differences.

Analysis of LCP (Figure 4IV) revealed that only Ag had a significantly higher value on the north wall. In contrast, Co, Mc, Tj, and Vm had higher LCP on the south wall, with no significant differences for the other five species. For east–west comparisons, Ls was higher on the east wall, whereas Ag, Lj, and Rp were higher on the west wall; no significant differences were observed for the other six species.

Regarding LSP (Figure 4V), Ag and Tj exhibited higher values on the north wall, while Vm was higher on the south wall. The other seven species showed no significant differences. In the east–west comparison, Ag and Fj had higher LSP on the west wall, with no significant differences among the remaining eight species.

Finally, analysis of R_d_ (Figure 4III) indicated that Ag had higher dark respiration on the north wall, whereas Co, Fj, Mc, Tj, and Vm were higher on the south wall. No significant differences were observed among the remaining four species. In the east–west comparison, only Ag showed lower R_d_ on the east wall; the other species exhibited no significant differences.

As shown in Figure 5, total biomass was highest for Ri, followed by Ag, then Mc and Rp, with Tj and Vm exhibiting the lowest values. Among these, the biomass of Ri and Mc did not differ significantly across the four wall orientations, indicating relatively uniform distribution. In contrast, Ag showed the highest biomass on the south wall, which was significantly higher than on the east and west walls and extremely significantly higher than on the north wall. Fj biomass was lowest on the south wall, being extremely significantly lower than on the east wall.

For total root-to-shoot ratio (RSR), Ag, Fj, and Ri exhibited the highest values, followed by Rp and Ls, whereas Co, Lj, Mc, Tj, and Vm had the lowest values. Ag displayed a significantly lower RSR on the south wall compared to the other three walls, among which differences were not significant. Fj showed considerable variation across walls, with the east wall value being significantly higher than the west wall and extremely significantly higher than the north and south walls. Ri reached its maximum RSR on the east wall, significantly higher than on the other walls. Mc had a significantly higher RSR on the south wall compared to the other walls. Lj and Ls followed a consistent pattern, being significantly lower on the north wall than on the other walls. Conversely, Vm exhibited the highest RSR on the north wall. The variation in Rp and Tj across the four walls was not significant.

### 2.4. PCA Results of Photosynthetic Parameter Variation Rates in 10 Green Wall Plants Across Different Orientations

To further explore differences in photosynthetic parameters among 10 green wall plant species across different wall orientations, principal component analysis (PCA) was conducted on the variation rates of these parameters, as summarized in Table 1. Two principal components were extracted: the first component, accounting for 45.33% of the variance, was primarily associated with dark respiration rate (R_d_) and light saturation point (LSP); the second component, accounting for 30.24% of the variance, was primarily associated with apparent quantum efficiency (AQE), maximum photosynthetic rate (P_max_), and light compensation point (LCP), with a cumulative explained variance of 75.57%.

Among the species, Ag, Co, Fj, Mc, Tj, and Vm exhibited positive comprehensive variation scores, whereas the remaining four species showed negative scores, indicating contrasting adaptive strategies in response to differing light environments.

As shown in Figure 6, cluster analysis based on the two principal components grouped the 10 plant species into three distinct categories. The first category consisted of the two monocotyledonous herbaceous species, Ag and Co. The second category included four dicotyledonous herbaceous and vine species, Fj, Mc, Tj, and Vm. The third category comprised four woody species, Lj, Ls, Ri, and Rp.

As shown in Table 2, the apparent quantum efficiency (AQE) of the first plant category was significantly less affected by changes in photosynthetically active radiation (PAR) than that of the second category. The maximum photosynthetic rate (P_max_) of the three categories did not differ significantly in response to PAR variations. The light compensation point (LCP) of the first category was significantly more sensitive to PAR changes than those of the second and third categories. Dark respiration rates (Rd) and light saturation point (LSP) of the first and second herbaceous categories were significantly more affected by PAR variations than those of the third woody category.

Overall, photosynthetic parameters of herbaceous plants in the first and second categories exhibited significantly greater sensitivity to PAR fluctuations than those of woody plants in the third category. Specifically, woody plants in the third category showed 45.8% lower variability in LCP and 64.5% lower variability in LSP compared to herbaceous plants in the first category.

### 2.5. “Photosensitivity-Growth Type-Ecological Function” Correlation Model and Plant Configuration

Using the coefficient of variation (CV = SD/Mean) of photosynthetic parameters as a quantitative metric, key principal components were extracted via PCA to establish a grading standard for plant light adaptation capacity (Table 2, Figure 6). Factor 1, accounting for 45.33% of the variance, was primarily associated with dark respiration rate (R_d_) and light saturation point (LSP). Negative Factor 1 values for woody species (Lj, Ls, Rp, Ri; ranging from −0.478 to −1.402) indicated high stability of R_d_ and LSP. Factor 2, accounting for 30.24% of the variance, was associated with apparent quantum efficiency (AQY), maximum photosynthetic rate (P_max_), and light compensation point (LCP). The negative Factor 2 value (−0.647) for the shade-tolerant herbaceous species Ag indicated stable AQY under low-light conditions.

Cluster analysis of the 10 plant species (Figure 5) revealed distinct patterns of photosynthetic variability. Monocotyledonous herbaceous plants (Ag and Co) exhibited high variability with mean CV values > 0.35. Specifically, Ag on the north wall showed a 41% increase in LCP variability, indicating high photosensitivity and limited low-light adaptability. Dicotyledonous herbaceous and vine species (Fj, Mc, Tj, Vm) exhibited moderate variability (mean CV 0.2–0.35), demonstrating strong light plasticity. Woody species (Lj, Ls, Rp, Ri) exhibited low variability (mean CV < 0.2), with photosynthetic parameter variability 45.8–64.5% lower than that of herbaceous species, reflecting outstanding photosynthetic stability.

Based on light gradient patterns and species-specific light adaptation traits, a three-dimensional planting configuration strategy is proposed (Table 3). Woody species maintain stable biomass across wall orientations, with root-to-shoot ratios reaching up to 0.51, serving as skeleton-forming species that enhance the structural stability of green wall vegetation. Climbing vines such as Mc fill gaps in the woody canopy layer on south-facing walls, while shade-tolerant herbaceous species such as Ag optimize coverage on north-facing walls under low-light conditions. This configuration enhances ecological functions and reduces maintenance demands, including irrigation requirements.

## 3. Discussion

### 3.1. Structural and Biochemical Basis of Light Adaptation Mechanisms

The regulation of plant photosynthetic characteristics by the light environment stems from the synergistic response of anatomical structure and biochemical processes. This study quantified the variation in photosynthetic parameters (such as LSP, LCP, and AQY) of 10 green wall plant species under the four-directional light gradient of a building facade, revealing the species-specific differences in light adaptation thresholds and their underlying mechanisms. Woody plants (e.g., Lj, Ls, Rp, and Ri) exhibited lower variation rates (CV < 0.2) in photosynthetic parameters. Their stability originates from structural adaptations: a thick cuticle (>8 μm) and a high proportion of lignified vessels (>30%) effectively reduce water transpiration loss and maintain stomatal conductance stability [21]. Conversely, the high variation rate (CV > 0.35) in herbaceous plants (e.g., Ag, Co) is related to thin mesophyll cells (<50 μm), leading to accelerated photorespiration under strong light and photosynthetic rate fluctuations (e.g., the variation rate of Vm on the south wall reached 0.516). At the biochemical level, sun plants (e.g., *Muehlenbeckia complexa*, Mc) under strong light efficiently scavenge reactive oxygen species through highly active antioxidant enzymes (SOD, POD, CAT), protecting the PSII reaction center [22], and likely rely on the PSII repair cycle (e.g., Deg- and FtsH protease-mediated D1 protein turnover) to maintain photochemical efficiency [23]. In contrast, shade plants (e.g., *Acorus gramineus* ‘Ogon’, Ag) maximize light capture efficiency through chloroplast light-avoidance movement [24], corresponding to a 41% increase in their LCP on the north wall. These mechanisms confirm that light adaptation is a structural–biochemical synergistic strategy, but the current research remains limited to phenotypic description. Future work needs to integrate microscopic evidence structurally, by using scanning electron microscopy (SEM) to quantify palisade tissue thickness (e.g., potentially 150–200 μm in Mc) and vein density (>8 mm/mm^2^) to analyze optimized light energy conduction [25], and biochemically, by using HPLC and ELISA techniques to track antioxidant enzyme dynamics and osmotic adjustment substance (e.g., proline) responses, combined with chlorophyll fluorescence imaging to visualize NPQ heterogeneity [26]. By establishing regression models between LCP, LSP, and microscopic indicators, the light signal transduction pathways (e.g., the upregulation mechanism of Lhcb proteins [27]) can be elucidated, providing theoretical targets for breeding high-light-efficiency plants.

### 3.2. Species-Specific Light Adaptation and Ecological Applications

Based on PCA clustering analysis, plant light adaptation strategies can be categorized into three types: woody plants (low variation type) and herbaceous/vine plants (medium-high variation type). This classification provides direct guidance for ecological applications. Woody plants (e.g., Rp, Ri) serve as a “structural skeleton” for green walls due to their low variation in photosynthetic parameters (45.8–64.5% lower than herbaceous plants), ensuring continuous coverage (canopy rate ≥ 95%) and significantly reducing maintenance costs (56% reduction in withered leaf pruning frequency) [8]. In terms of ecological benefits, the high root-to-shoot ratio (RSR = 0.51) of woody plants enhances water use efficiency, while vines (e.g., *Muehlenbeckia complexa*, Mc) contribute significantly to cooling through high transpiration rates (up to 12 μmol·m^−2^·s^−1^ on south walls, achieving a temperature reduction of 3.1 °C). The synergy between these plant types can reduce building cooling load by 18.3% [28]. For configuration strategies, south-facing walls are recommended to adopt “woody plants + sun-loving vines” (e.g., Lj + Mc) to maximize carbon sequestration (54% increase) and cooling; north-facing walls should utilize “shade-tolerant herbaceous plants” (e.g., Ag + Fj) to maintain 86% coverage under low light conditions while saving 32% water. However, moving beyond single photosynthetic indicators, it is necessary to integrate functional trait databases (e.g., specific leaf area SLA, wood density, leaf dry matter content LDMC) to assess resource utilization strategies [29]. For instance, woody plants with low SLA and high LDMC (e.g., *Ligustrum japonicum* ‘Howardii’, Lj) exhibit strong resistance to mechanical stress, making them more suitable for high-rise environments with wind pressure. Long-term monitoring of species’ survival rates, coverage stability, and response to low maintenance under root zone restriction can help filter truly “low-input, high-efficiency” species [11]. Meanwhile, exploring mixed planting with complementary ecological niches (e.g., combining vines Tj with shrubs Ls) leverages differences in canopy structure, phenology, and root systems to enhance biodiversity and ecological stability. The characteristic of root zone environmental constraints, which is equally important as the light gradient affecting the aboveground-underground synergistic growth of green wall plants, is also a key parameter for studying the improvement of the ecological efficiency of green walls. In production applications, module size, substrate characteristics, and irrigation systems are also important environmental factors that determine the configuration, distribution, and dynamic changes of plant root systems [30]. In the case of the Parkroyal project in Singapore, the 3-year survival rate of woody plants under the dual guarantee of a specifically structured deep substrate and intelligent irrigation is over 90%, whereas it drops to 70% otherwise [8]. Ultimately, it is essential to develop species configuration lists based on light gradient-functional trait matrices and establish life cycle cost–benefit analysis (LCCBA) models to promote the transition of photovoltaic-green systems from technical feasibility to economic and social benefits, supporting emission reduction goals in high-density cities.

### 3.3. Research Limitations and Future Directions

Although this study has revealed mechanisms of light adaptation, it still exhibits multidimensional limitations. Firstly, the spatial scale is singular, and conclusions need validation across different climate zones and urban forms to develop regionally adaptive models (for instance, arid regions require balancing water use efficiency WUE with photovoltaic cooling). Secondly, the regulatory role of rhizosphere microorganisms on light gradients has not been considered; they may modulate root water uptake through hormone signals, affecting the stability of woody plants [13]. Temporally, the lack of seasonal dynamic data limits annual performance evaluation, and the impacts of long-term succession, changes in substrate physicochemical properties (e.g., salt accumulation), and extreme climate events remain unquantified. Thirdly, the depth of system coupling is insufficient; interactions among “light-green-substrate” often remain at correlational levels, and intrinsic causal mechanisms are not clarified [31]. For system-level performance optimization, future research needs to construct a coupled heat transfer model of plant-substrate-building, analyze the relationship between plant transpiration (Penman-Monteith equation) and biomass through a full-scale environmental chamber experimental platform, and realize the unified quantification of energy–ecological output [32,33]. Regarding the evaluation and verification of the synergistic efficiency of light and greenery, this study failed to reveal the nonlinear coupling mechanism of photovoltaic shading–plant transpiration–building thermal inertia, and the relationship between the power generation efficiency gain brought by the reduction of photovoltaic panel temperature and plant transpiration water consumption (ET) has not yet been quantified. Future work should integrate root zone sensing technologies with IoT platforms to develop light-water intelligent algorithms [34,35], incorporating real-time monitoring and machine learning for dynamic irrigation optimization (based on LSP/LCP thresholds). Simultaneously, introduce isotopic labeling (e.g., ^15^N), molecular ecology (high-throughput sequencing), and CFD simulations to elucidate cross-scale mechanisms of energy-water-carbon-nutrient cycles [36]. At the application level, construct digital twin models for precise water and fertilizer management, and promote the inclusion of photosynthetic stability indicators into green building certifications, advancing NbS standardization [4]. Ultimately, precise design and ecological function enhancement of green walls can be achieved through a “light-plant-technology” closed loop.

## 4. Materials and Methods

### 4.1. Study Sites

As shown in Figure 7, the research was conducted at a commercial complex located at the entrance of the Shanghai Disney International Resort Ecological Park (121°40′–121°41′ E, 31°08′–31°09′ N) in Pudong New District, Shanghai. According to 2022 records from the Shanghai Meteorological Bureau, the area has a mean annual temperature of 16.5 °C and receives an average annual precipitation of 1064.1 mm. The mean relative humidity is 77%, and the mean wind speed is 2.5 m/s. The cumulative annual solar radiation is 4635.7 MJ/m^2^, with an annual sunshine duration of 1879.7 h. The green wall measures 27 m in height and 120 m in width, with a total area of 3240 m^2^.

### 4.2. Green Wall Systems and Plant Materials

The green wall was installed on the façade of a commercial complex building with multiple orientations, with an average height of 6 m and a total surface area of 3314 m^2^. The supporting framework consisted of galvanized steel components connected by bolts. Modular planting trays were made of polypropylene (PP), with individual pots measuring 450 mm in length, 100 mm in width, and 85 mm in height; three pots were grouped as a single unit, each fitted with an automated irrigation system. The substrate used was a standardized mixed medium (Traditional Soil Formulation Medium, TSFM), composed of earthworm soil, peat, and vermiculite in a 1:1:1 volume ratio. This granular substrate, widely applied in sowing, cutting propagation, and transplanting, was packed into the planting containers.

Ten ornamental plant species from different families, exhibiting diverse foliage traits (Table 4), were transplanted in July 2016 onto green walls oriented north, east, south, and west. Due to module design constraints, each square meter was planted with 39 seedlings, with all species established at equal density. Under uniform conditions of soil, irrigation, fertilization, and maintenance, three years of growth (by July 2019) resulted in clear orientation-dependent differences in plant performance. Each species developed a monoculture patch exceeding 20 m^2^ in area. In this study, no deliberate criteria were used by researchers or maintenance personnel to establish monoculture patches. The patterns arose naturally due to orientation-dependent environmental variations (e.g., light, temperature), which triggered species-specific growth responses and competitive interactions over the three-year period. For instance, south-facing walls favored light-loving species through an increased photosynthesis rate, while north-facing walls promoted shade-tolerant species via reduced photoinhibition. This self-organization underscores the role of microclimatic factors in shaping plant communities on green walls.

### 4.3. Experimental Design and Data Collection

All experiments were conducted on clear days in August 2019. Samples were randomly collected from four orientations within single-species communities covering more than 20 m^2^, at a height of 2–2.5 m above ground level. A consistent sampling height was maintained to assess the effects of orientation-specific photosynthetically active radiation (PAR) environments on photosynthesis. For each of the 10 plant species, we sampled 9 individual plants per orientation (north, east, south, west), resulting in a total of 36 plants per species across all directions. From each plant, 9 mature, undamaged leaves were measured for photosynthetic parameters, yielding n = 9 biological replicates per species per orientation. During the measurement period (6:00–18:00), a portable photosynthesis system (LI-6400XT, LI-COR, Lincoln, NE, USA) was used to record environmental variables including temperature, CO_2_ concentration, and PAR at the north-, east-, south-, and west-facing walls. Measurements were taken hourly, with three replicates, to characterize diurnal variation.

(1) Measurement and analysis of photosynthetic light-response curves: Nine intact, mature, and undamaged leaves were sampled from nine individuals of each species, all planted simultaneously with uniform specifications. Measurements were performed between 9:00 and 11:00 AM. Under an airflow rate of 500 μmol·s^−1^ and a reference CO_2_ concentration of 400 ppm, photosynthetic light-response curves were obtained at 16 photosynthetic photon flux densities, ranging from 2000 to 0 μmol·m^−2^·s^−1^ (2000, 1800, 1600, 1400, 1200, 1000, 800, 600, 400, 200, 150, 100, 80, 50, 20, 0). The instantaneous net photosynthetic rate (P_n_) of leaves was recorded. For each wall orientation, the average photosynthetic rate of the ten species was calculated under varying PAR levels. Photosynthesis rate across treatments were compared using a non-rectangular hyperbola model derived from nonlinear regression to estimate maximum photosynthetic rate (P_max_). As expected, shade-tolerant species typically exhibited lower P_max_ values than sun-loving species.

Linear regression of data within the 0–2000 μmol·m^−2^·s^−1^ range was used to calculate the apparent quantum yield (AQY), represented by the slope of the initial linear portion of the curve, which reflects the efficiency of light energy utilization under low light. At zero PAR, the y-intercept corresponds to the dark respiration rate (R_d_). The light compensation point (LCP) was calculated as the ratio of R_d_ to AQY, corresponding to the *x*-axis intercept of the light-response curve; it represents the light intensity at which photosynthetic carbon gain balances respiratory carbon loss. The light saturation point (LSP) was defined as the ratio of the sum of P_max_ and R_d_ to AQY, reflecting the photosynthetic photon flux density (PPFD) at which photosynthesis reaches saturation. A higher LSP indicates greater tolerance to high irradiance and reduced susceptibility to photoinhibition under strong light.P_n_ = {AQY × PAR + P_max_ − [(AQY × PAR + P_max_)2 − 4 × k × AQY × PAR × P_max_] − 2}/(2 × k) − R_d_(1)LCP = R_d_/AQY(2)(3)$$LSP=(R_{d}+P_{max})/AQY$$
where AQY denotes the apparent quantum yield, defined as the initial slope of the light-response curve and representing the efficiency with which plants fix CO_2_ per photon absorbed under low-light conditions. PAR refers to photosynthetically active radiation, i.e., the spectral range of solar radiation (400–700 nm) utilized for photosynthesis. P_max_ indicates the maximum photosynthetic rate, representing the highest rate achieved under saturating light. R_d_ signifies the dark respiration rate, referring to the rate of CO_2_ release by plants in the absence of light. LCP denotes the light compensation point, defined as the light intensity at which photosynthetic carbon assimilation equals respiratory carbon loss. LSP denotes the light saturation point, representing the irradiance level at which photosynthetic rate reaches saturation.

The coefficient of variation (CV) is commonly used to assess the variability of photosynthetic parameters within a single plant species across green walls with different orientations. It is calculated as the ratio of the standard deviation (SD) to the mean. Higher CV values indicate greater variability in photosynthetic parameters under differing photosynthetically active radiation (PAR) conditions, reflecting a stronger influence of the PAR environment.CV = SD/Mean(4)

Biomass Measurement: After measuring the photosynthetic parameters, plants were cleaned, and their organs separated. The plant tissues were first killed at 105 °C for 2 h, then oven-dried at 75 °C for 3 days until a constant weight was reached. Dry weights of leaves (Leaf Dry Weight, LDW), aboveground biomass (Above-Ground Biomass, AGB), and root biomass (Root Biomass, RB) were recorded using a balance. Specific leaf area (SLA) and root-to-shoot ratio (RSR) were subsequently calculated to reflect the growth status and health of each plant species on different wall orientations.RSR = RB/AGB(5)

### 4.4. Statistical Analyses

In this study, figures and tables were prepared using Microsoft Excel 2016, and Data were analyzed using SPSS 23 software. For comparisons of photosynthetic parameters (AQE, P_max_, LCP, LSP, R_d_) across the four orientations for each species, we used one-way ANOVA followed by Tukey’s HSD post hoc test for multiple comparisons. Significance was set at *p* < 0.05. Prior to ANOVA, data were checked for normality using the Shapiro–Wilk test and for homogeneity of variances using Levene’s test. If assumptions were violated, non-parametric tests such as Kruskal–Wallis with Dunn’s post hoc test were applied. Biomass and root-to-shoot ratio (RSR) of 10 plant species were quantified to assess differences among treatments. Principal component analysis (PCA) was applied to the variation rates of photosynthetic parameters to extract significant components, followed by hierarchical clustering to analyze patterns systematically.

## 5. Conclusions

This study systematically quantified the effects of four-directional light gradients on the photosynthetic adaptability of 10 green wall plant species through three years of in situ monitoring. Key findings include the following:

(1) Pronounced photosynthetic homeostasis in woody plants: Shrubs (e.g., Lj, Ls, Rp, Ri) exhibited 45.8–64.5% lower variability in photosynthetic parameters compared to herbaceous species. Their stable morphogenesis and enhanced photosynthetic performance substantially improved long-term green wall coverage, challenging the conventional notion that “herbaceous plants possess higher plasticity”.

(2) Species-specific light adaptation strategies: The LSP of Mc on the south-facing wall is 32% higher than that on the north-facing wall, and the LCP of Ag on the north-facing wall is 41% lower than that on the south-facing wall. Dicotyledonous herbs and vines displayed significantly higher light sensitivity than woody plants. These patterns provide a physiological basis for precision plant configuration.

(3) Configuration principle of “woody plants as the framework, complemented by vines and herbs”: Woody species serve as the core community-building elements for green walls. By integrating plant light adaptation standards, functional plant community configuration models were developed:South-facing walls: Replacing traditional Crassula species with Mc vines, combined with a Lj woody framework, enhances summer cooling efficiency.North-facing walls: A cost-effective solution employs an Ag + Fj combination, maintaining 86% coverage under low-light conditions (<500 μmol·m^−2^·s^−1^) and achieving 32% water savings compared to pure herbaceous systems, with a root-to-crown ratio as high as 0.51.East- and west-facing walls: Mixed woody-vine configurations are recommended to balance carbon sequestration and shading benefits.

Based on the photosynthetic adaptation data systematically collected in this study, the strategic alignment of plant species with their optimal microenvironments significantly enhances vertical greening quality and efficiency. This trait-environment matching framework ensures robust ecological performance, sustainable resource utilization, and long-term stability in high-density urban settings. Future research should further investigate plant growth regulation under root-restricted conditions and develop novel substrate materials tailored to urban habitats. Such advancements would enhance plant adaptability, substantially reduce maintenance frequency, and lower replanting costs.

## Figures and Tables

**Figure 1 plants-14-03570-f001:**
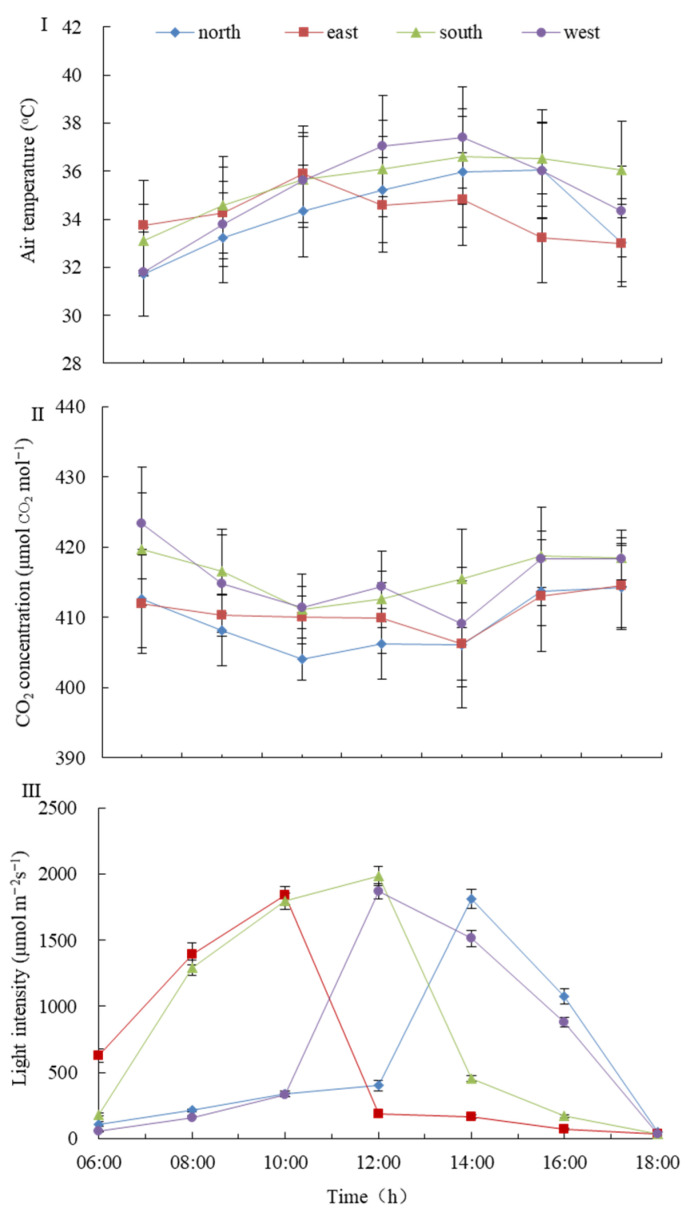
Daily variation in atmospheric factors for the 4 aspects of the green wall. Notes: (**I**) Diurnal variations in air temperature for four cardinal directions (North, East, South, West); (**II**) Changes in atmospheric CO_2_ concentration (μmol CO_2_ mol^−1^) recorded simultaneously from different orientations; (**III**) Photosynthetically active radiation (PAR) intensity (μmol m^−2^ s^−1^) monitored across the four directions throughout the day. All measurements were conducted using a portable photosynthesis system (LI-6400XT, LI-COR, USA) with data presented as mean ± SD (n = 3). In the legend, north, east, south, and west represent the four corresponding orientations of the green wall.

**Figure 2 plants-14-03570-f002:**
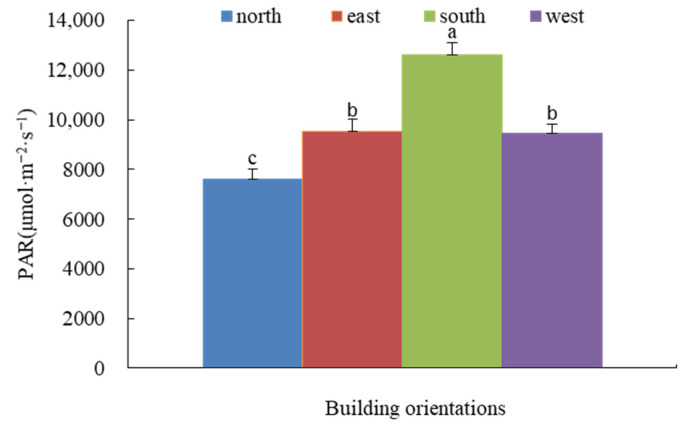
Daily photosynthetically active radiation for the 4 aspects of the green wall. Notes: In the legend, north, east, south, and west represent the four corresponding orientations of the green wall. The abbreviations of the parameters on the ordinate in the figure are as follows, PAR represents photosynthetically active radiation. do not differ significantly (*p* < 0.05). Error bars represent ± SD. Bars labeled with different letters (a, b, or c) indicate significant differences between orientations for photosynthetically active radiation (PAR) (one-way ANOVA, Tukey’s HSD, *p* < 0.05).

**Figure 3 plants-14-03570-f003:**
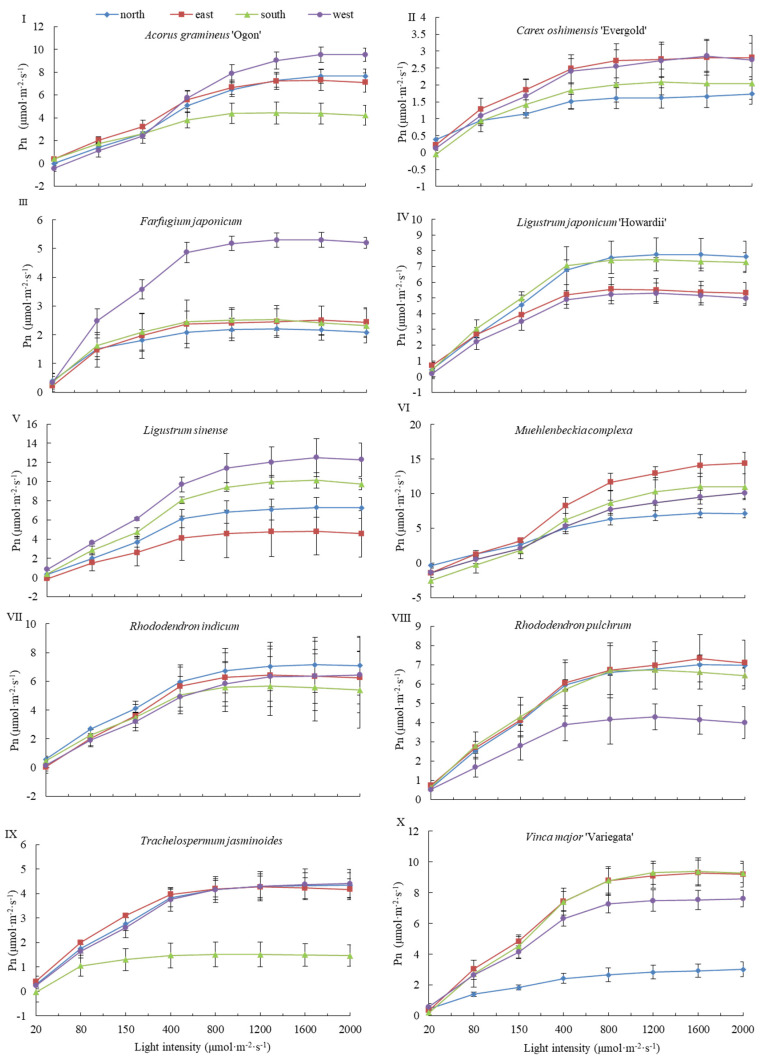
Photoresponse curves of 10 species on the green wall from 4 aspects (±SD, N = 9). Notes: (**I**–**X**) Photosynthetic light response curves for *Acorus gramineus* ‘Ogon’, *Carex oshimensis* ‘Evergold’, *Farfugium japonicum*, *Ligustrum japonicum* ‘Howardii’, *Ligustrum sinense*, *Muehlenbeckia complexa*, *Rhododendron* × *pulchrum*, *Rhododendron indicum*, *Trachelospermum jasminoides* ‘Flame’ and *Vinca major* ‘Variegata’ green walls with four orientations The abbreviations of the parameters on the ordinate in the figure are as follows, Pn represents the instantaneous net photosynthetic rate. In the legend, north, east, south, and west represent the four corresponding orientations of the green wall.

**Figure 4 plants-14-03570-f004:**
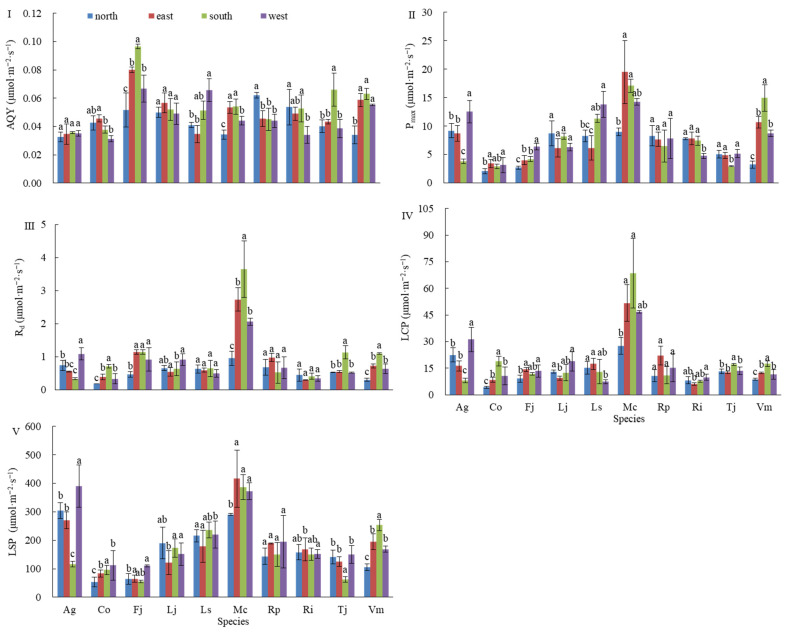
Mean (±SD, N = 9) of photosynthetic parameters of 10 species on the green wall from 4 aspects. Notes: (**I**) Apparent quantum yield (AQY) values for 10 species under four orientations. (**II**) Maximum photosynthetic rate (Pmax) variation across north, east, south, and west exposures. (**III**) Dark respiration rate (Rd) measurements under light-free conditions. (**IV**) Light compensation point (LCP) thresholds for each plant species. (**V**) Light saturation point (LSP) levels indicating photosynthetic plateau. Data are presented as mean ± SD (n = 9). Different letters indicate significant differences between orientations for each species (one-way ANOVA, Tukey’s HSD, *p* < 0.05). In the legend, north, east, south, and west represent the four corresponding orientations of the green wall. The abbreviations of the parameters on the ordinate in the figure are as follows, AQY represents apparent quantum yield; P_max_ represents the maximum photosynthesis rates; R_d_ represents the dark respiration rate; LCP represents light compensation point; LSP represents light saturation point. The abbreviations for the horizontal axis parameters in the figure are as follows: Ag represents *Acorus gramineus* ‘Ogon’; Co represents *Carex oshimensis* ‘Evergold’; Fj represents *Farfugium japonicum*; Lj represents *Ligustrum japonicum* ‘Howardii’; Ls represents *Ligustrum sinense*; Mc represents *Muehlenbeckia complexa*; Rp represents *Rhododendron* × *pulchrum*; Ri represents *Rhododendron indicum*; Tj represents *Trachelospermum jasminoides* ‘Flame’; Vm represents *Vinca major* ‘Variegata’.

**Figure 5 plants-14-03570-f005:**
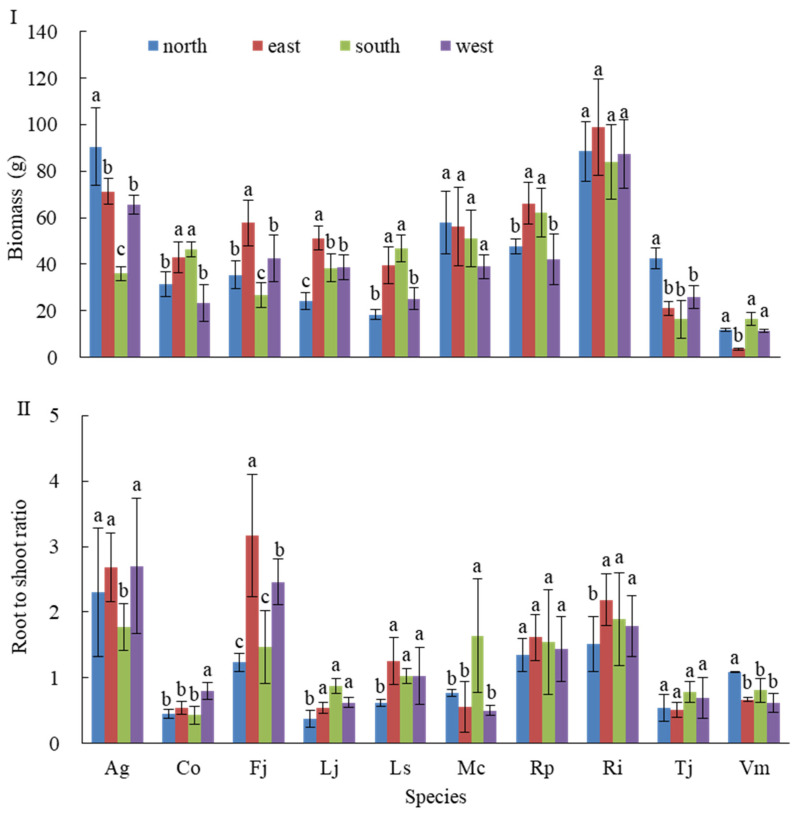
Total biomass and root-to-shoot ratio of different species on the green wall from 4 aspects (±SD, N = 9). Notes: (**I**) Total biomass accumulation for 10 plant species across the four wall orientations; (**II**) Root-to-shoot ratio (RSR) values reflecting resource allocation strategies. Data are presented as mean ± SD (n = 9). Different letters indicate significant differences between orientations for each species (one-way ANOVA, Tukey’s HSD, *p* < 0.05). In the legend, north, east, south, and west represent the four corresponding orientations of the green wall. The abbreviations for the horizontal axis parameters in the figure are as follows: Ag represents *Acorus gramineus* ‘Ogon’; Co represents *Carex oshimensis* ‘Evergold’; Fj represents *Farfugium japonicum*; Lj represents *Ligustrum japonicum* ‘Howardii’; Ls represents *Ligustrum sinense*; Mc represents *Muehlenbeckia complexa*; Rp represents *Rhododendron* × *pulchrum*; Ri represents *Rhododendron indicum*; Tj represents *Trachelospermum jasminoides* ‘Flame’; Vm represents *Vinca major* ‘Variegata’.

**Figure 6 plants-14-03570-f006:**
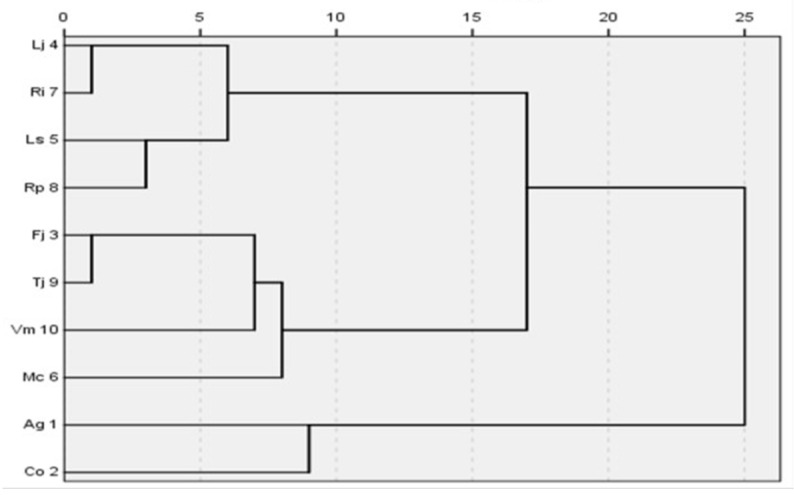
Clustering analysis based on the coefficient of variation in the photosynthetic parameters for the 10 species on the green wall. Notes: The abbreviations of the parameters in the figure are as follows, Ag represents *Acorus gramineus* ‘Ogon’; Co represents *Carex oshimensis* ‘Evergold’; Fj represents *Farfugium japonicum*; Lj represents *Ligustrum japonicum* ‘Howardii’; Ls represents *Ligustrum sinense*; Mc represents *Muehlenbeckia complexa*; Rp represents *Rhododendron* × *pulchrum*; Ri represents *Rhododendron indicum*; Tj represents *Trachelospermum jasminoides* ‘Flame’;Vm represents *Vinca major* ‘Variegata’.

**Figure 7 plants-14-03570-f007:**
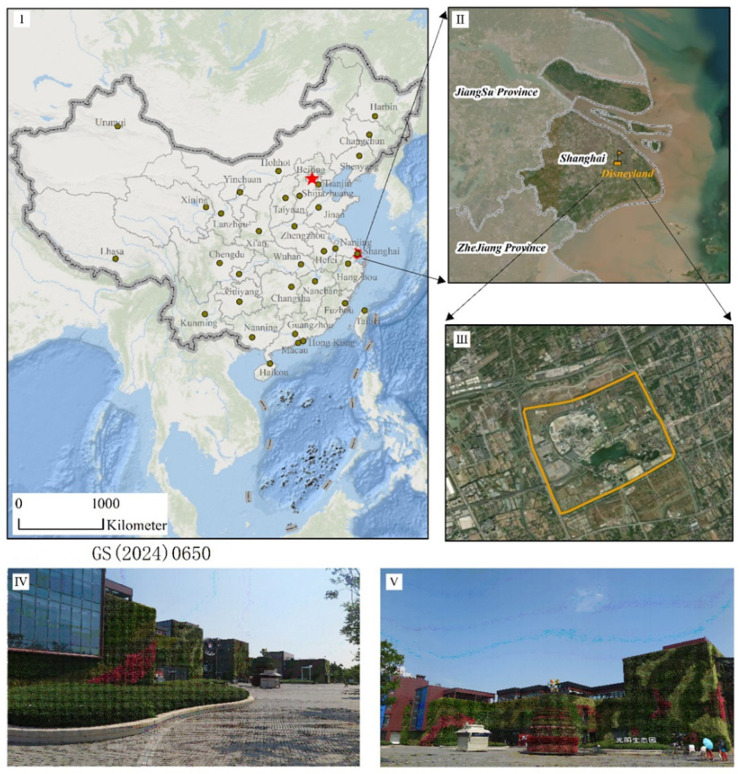
The exact location of the green wall mentioned in the test is Shanghai in eastern China. Notes: (**I**–**III**) shows the location of Disneyland in Shanghai, China. (**IV**,**V**) shows the figure of commercial complex green wall.

**Table 1 plants-14-03570-t001:** PCA of the photosynthetic parameters for the 10 species on the green wall.

Species	The Coefficient of Variation of Photosynthetic Parameters					Component		
	AQY	P_max_	R_d_	LCP	LSP	Factor_1_	Factor_2_	Factor
Ag	0.037	0.420	0.472	0.504	0.424	1.781	−0.647	1.285
Co	0.159	0.205	0.557	0.586	0.286	1.071	−1.073	0.435
Fj	0.259	0.358	0.350	0.186	0.338	0.006	1.254	0.625
Lj	0.066	0.179	0.229	0.295	0.187	−0.478	−1.182	−1.015
Ls	0.281	0.341	0.117	0.325	0.112	−1.081	0.431	−0.761
Mc	0.199	0.301	0.484	0.346	0.147	0.088	−0.042	0.058
Rp	0.193	0.211	0.185	0.200	0.050	−1.402	−0.253	−1.388
Ri	0.175	0.102	0.270	0.359	0.154	−0.725	−0.996	−1.145
Tj	0.272	0.234	0.443	0.145	0.328	−0.133	1.061	0.405
Vm	0.244	0.516	0.479	0.293	0.341	0.874	1.446	1.501

Notes: The abbreviations of the parameters in the table are as follows, AQY represents apparent quantum yield; P_max_ represents the maximum photosynthesis rates; R_d_ represents the dark respiration rate; LCP represents light compensation point; LSP represents light saturation point. Ag represents *Acorus gramineus* ‘Ogon’; Co represents *Carex oshimensis* ‘Evergold’; Fj represents *Farfugium japonicum*; Lj represents *Ligustrum japonicum* ‘Howardii’; Ls represents *Ligustrum sinense*; Mc represents *Muehlenbeckia complexa*; Rp represents *Rhododendron* × *pulchrum*; Ri represents *Rhododendron indicum*; Tj represents *Trachelospermum jasminoides* ‘Flame’; Vm represents *Vinca major* ‘Variegata’.

**Table 2 plants-14-03570-t002:** Mean (the standard deviation, ±SD) and their differences in the coefficient of variation of photosynthetic parameters for the 3 types of species on the green wall.

Classification	AQY (μmol·m^−2^·s^−1^)	P_max_ (μmol·m^−2^·s^−1^)	R_d_ (μmol·m^−2^·s^−1^)	LCP (μmol·m^−2^·s^−1^)	LSP (μmol·m^−2^·s^−1^)	Factor
I	0.098 ± 0.086 b	0.313 ± 0.152 a	0.515 ± 0.060 a	0.545 ± 0.058 a	0.355 ± 0.098 a	0.860 ± 0.601 a
II	0.244 ± 0.032 a	0.352 ± 0.120 a	0.439 ± 0.062 a	0.243 ± 0.093 b	0.289 ± 0.094 a	0.647 ± 0.615 a
III	0.179 ± 0.088 ab	0.208 ± 0.100 a	0.200 ± 0.065 b	0.295 ± 0.068 b	0.126 ± 0.059 b	1.077 ± 0.261 b
F	4.289	1.817	28.114	15.884	8.608	19.909
*p*	0.054	0.224	<0.001	0.002	0.001	0.001

Notes: I is for two species of monocotyledonous herbs; II is for dicotyledonous herbs and lianas; III is for woody plants. The abbreviations of the parameters in the table are as follows, AQY represents apparent quantum yield; P_max_ represents the maximum photosynthesis rates; R_d_ represents the dark respiration rate; LCP represents light compensation point; LSP represents light saturation point. Different lowercase letters (a, b) in the same column indicate significant differences among the three groups (Classification I, II, III) at the 0.05 significance level (one-way ANOVA, Tukey’s HSD, *p* < 0.05).

**Table 3 plants-14-03570-t003:** The model for plant configuration in the green wall is the “growth type—orientation” functional matrix.

Building Orientation	Lighting Environment Characteristics	Recommended Plant Combinations	Ecological Function Advantages
South Wall	Intense light (PAR peak 1500 μmol·m^−2^·s^−1^, duration 5 h)	Woody framework + sun-loving vines (Lj/Ls + Mc)	Carbon sequestration (Mc—LSP ↑32%) + Cooling effect (Transpiration rate 12 μmol·m^−2^·s^−1^)
East Wall	Morning high-intensity light (PAR > 1500 μmol·m^−2^·s^−1^, 3 h)	Woody + Vigorously Climbing Vine (Rp + Vm)	Shading Effectiveness (Vm: Canopy Cover 86%)
West Wall	Afternoon high light (PAR > 1500 μmol·m^−2^·s^−1^, 2 h)	Woody + Drought-tolerant herbaceous plants (Ri + Fj)	Increase water use efficiency (Fj root-to-shoot ratio 0.51)
North Wall	Low light (PAR peak < 800 μmol·m^−2^·s^−1^)	Shade-loving herbaceous plants (Ag + Co)	Low maintenance (Water savings 32%+; Replanting costs ↓ 73%)

Notes: The abbreviations of the parameters in the table are as follows, Ag represents *Acorus gramineus* ‘Ogon’; Co represents *Carex oshimensis* ‘Evergold’; Fj represents *Farfugium japonicum*; Lj represents *Ligustrum japonicum* ‘Howardii’; Ls represents *Ligustrum sinense*; Mc represents *Muehlenbeckia complexa*; Rp represents *Rhododendron* × *pulchrum*; Ri represents *Rhododendron indicum*; Vm represents *Vinca major* ‘Variegata’. PAR represents photosynthetically active radiation; LSP represents light saturation point.

**Table 4 plants-14-03570-t004:** The ten plant species (and abbreviations) used in this experiment and Family, light requirements, Seedling specifications, planting area and density of the green wall systems ([7,37,38,39]).

Serial Number	Latin Name	Abbreviation	Seedling Specifications	Planting Area (m^2^)	Image	Photoperiodism/Growth Type
1	*Acorus gramineus* ‘Ogon’	Ag	Crown width: 25 cm, Height: 25 cm, 4 buds	389.34	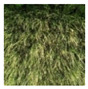	shade-tolerant/herbaceous
2	*Carex oshimensis* ‘Evergold’	Co	Crown width: 20 cm, height: 15 cm, 4 buds	183.39	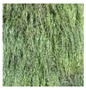	light-demanding/herbaceous
3	*Farfugium japonicum*	Fj	Crown width: 25 cm, height: 15 cm, with 7 or more leaves	288.13	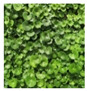	intermediate/herbaceous
4	*Ligustrum japonicum* ‘Howardii’	Lj	Crown width: 20 cm, height: 25 cm	216.71	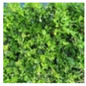	intermediate/woody
5	*Ligustrum sinense*	Ls	Crown width: 20 cm, height: 15 cm	150.52	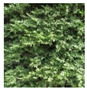	intermediate/woody
6	*Muehlenbeckia complexa*	Mc	Crown width: 25 cm, height: 5–10 cm, Lush foliage	270.33	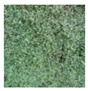	light-demanding/liana
7	*Rhododendron × pulchrum*	Rp	Crown width: 25 cm, height: 20–25 cm	249.19	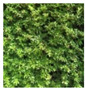	intermediate/woody
8	*Rhododendron indicum*	Ri	Crown width: 25 cm, height: 20–25 cm	270.29	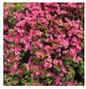	intermediate/woody
9	*Trachelospermum jasminoides* ‘Flame’	Tj	Crown width: 25 cm, height: 5–10 cm, Branch 7	260	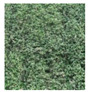	intermediate/liana
10	*Vinca major* ‘Variegata’	Vm	Crown width: 20 cm, height: 20 cm, Branch 7	584.42	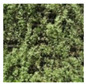	intermediate/liana

Notes: 48–60 plants per square meter, with the actual number determined by achieving full coverage.

## Data Availability

The original contributions presented in this study are included in the article. Further inquiries can be directed to the corresponding authors.

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
