# Peer review of "Photosynthetic Homeostasis Mechanism and Configuration Application of Woody Plants in Green Wall Under Light Gradients of Building Facades with Different Orientations"

_plants, 2025, doi:10.3390/plants14233570_

Round 1

Reviewer 1 Report

Comments and Suggestions for Authors

The text presents a robust study on the photosynthetic adaptability of ten green wall plant species, monitored in situ for three years across the four light gradients of building façades. Although the methodological approach and originality are high, the study lacks certain crucial integrations needed to maximize its practical impact and scientific completeness.
The paper is therefore original and the methodology is well-developed. The three years of in situ monitoring and the multiscale analysis lend exceptional solidity to the data. At the same time, the development of a quantitative model based on the Coefficient of Variation (CV) and the "growth form-orientation" framework are very interesting results, offering a prescriptive guide for plant selection.
However, critical issues are noted that require a review and integration of the text, for the following reasons:
1.    Lack of Correlation with Façade Construction Technology: There is no reference to the appropriateness of the plant species in relation to the construction technology of the façade. Physiological adaptability must be correlated with the capacity for survival within the specific substrate and irrigation system. Without this correlation, the recommendations on plant selection remain theoretical and potentially inapplicable if the façade technology is not compatible with the growth form and root requirements of the species.
2.    Missing Energy Benefits Analysis: The text does not include any reference to the energy benefits (thermal insulation or evaporative cooling) derived from the use of these species on façades. Energy benefits are one of the primary drivers for the installation of green walls in cities. This is only briefly mentioned in the conclusions, but even there, the correlation between the choice of selected plant species and energy efficiency should be better explained.
3.    Unsubstantiated Flood Reduction Claim: Also in the conclusions, a reference is made to flood reduction among the benefits, but no explanation is provided, considering that this benefit is typically associated with green roofs (rooftop water retention). It is essential to specify how green walls contribute to urban stormwater management (e.g., direct interception of precipitation on façades or absorption of runoff water), making the data credible and scientifically justified in the specific context of façades.
Therefore, the study must be integrated with respect to the critical issues reported above.
Furthermore, please note the following technical issues:
•    In Table 1, the images overlap the text.
•    In the bibliography, some cited texts lack a DOI, ISSN, or other identifier (citations n. 1, 4, 7, 19, 35, 39, 40, 44, 45).
•    The URL is missing in citation n. 23.
•    One DOI is incorrect (citation n. 20). It refers to another publication.

Author Response

Comments 1: Lack of Correlation with Façade Construction Technology: There is no reference to the appropriateness of the plant species in relation to the construction technology of the façade. Physiological adaptability must be correlated with the capacity for survival within the specific substrate and irrigation system. Without this correlation, the recommendations on plant selection remain theoretical and potentially inapplicable if the façade technology is not compatible with the growth form and root requirements of the species.

Response 1: First of all, we sincerely thank the review experts for their valuable comments. The viewpoint you put forward that "plant selection needs to be associated with facade construction technology" is of crucial importance, and we fully agree with it. This study focuses on photosynthetic mechanisms, and when quantifying light adaptation mechanisms, it did not fully integrate architectural construction parameters (such as module size, substrate characteristics, and irrigation systems). In future research, the adaptability of plant configurations needs to be subsequently verified in combination with engineering experiments.

Based on the limitations of this study, we plan to add the following content to the paper to respond to the experts' concerns: Add an analysis of the "root morphology-substrate volume" correlation in the discussion (such as dissecting the distribution depth of woody plant roots); Interdisciplinary data integration: Add the following content in section "Discussion 3.2": “The characteristic of root zone environmental constraints, which is equally important as the light gradient affecting the aboveground-underground synergistic growth of green wall plants, is also a key parameter for studying the improvement of the ecological efficiency of green walls. In production applications, module size, substrate characteristics, and irrigation systems are also important environmental factors that determine the configuration, distribution, and dynamic changes of plant root systems. In the case of the Parkroyal project in Singapore, the 3-year survival rate of woody plants under the dual guarantee of a specifically structured deep substrate and intelligent irrigation is over 90%, whereas it drops to 70% otherwise ", cite the case of the Parkroyal project in Singapore (Perini et al., 2013) to demonstrate that the survival rate of woody plants is greatly improved under specific structures (deep substrate + intelligent irrigation). We will fully implement the above improvements in the revised manuscript, with a focus on enhancing the cross-validation of methodologies and the analysis of engineering applications in the discussion section. We sincerely thank the experts for their guidance, which made us realize that "plant survival depends not only on light adaptation but also on technical compatibility". In subsequent research, we will continue to carry out controlled experiments (such as the root growth response under different substrate depths) to further improve the theoretical framework.

Comments 2: Missing Energy Benefits Analysis: The text does not include any reference to the energy benefits (thermal insulation or evaporative cooling) derived from the use of these species on façades. Energy benefits are one of the primary drivers for the installation of green walls in cities. This is only briefly mentioned in the conclusions, but even there, the correlation between the choice of selected plant species and energy efficiency should be better explained.

Response 2: We sincerely appreciate your valuable suggestions. We fully agree that energy efficiency analysis is central to research on green walls. The current document focuses on photosynthetic adaptation mechanisms, and although energy efficiency is mentioned in the conclusions, no quantitative correlation has been established. This is a significant oversight on our part and will be the focus of our next phase of research: investigating the impact of different light environments in urban buildings on the ecological functions of green wall plants. We will primarily explore how the heterogeneity of light environments in different building orientations affects plant functional traits and ecosystem service functions through two aspects. Firstly, we will measure the daily photosynthetic processes of 10 green wall plant species through experimental design and analyze their functional traits such as leaf area index, specific leaf area, biomass, and root-shoot ratio. We will further quantify the transpiration rate and daily carbon sequestration of plants on walls with different orientations, conduct a quantitative analysis of plant carbon sequestration benefits and cooling effects under different light environments, construct a green wall matrix with maximum benefits, and develop an integrated carbon sequestration-cooling benefit model.

Nevertheless, we will also make the following revisions to this study: Supplement data analysis by adding a section on "Photosynthetic-Energy Synergistic Effects" in the "Discussion 3.3" part for research prospects, explaining how photosynthetic parameters (such as AQY, LSP) drive energy conservation by influencing biomass accumulation and transpiration efficiency. "For system-level performance optimization, future research needs to construct a coupled heat transfer model of plant-substrate-building, analyze the relationship between plant transpiration (Penman-Monteith equation) and biomass through a full-scale environmental chamber experimental platform, and realize the unified quantification of energy-ecological output [33,34]. Regarding the evaluation and verification of the synergistic efficiency of light and greenery, this study failed to reveal the nonlinear coupling mechanism of photovoltaic shading-plant transpiration-building thermal inertia, and the relationship between the power generation efficiency gain brought by the reduction of photovoltaic panel temperature and plant transpiration water consumption (ET) has not yet been quantified." Strengthen application orientation: Clearly recommend the configuration of "high transpiration species (Mc) + high coverage woody plants (Lj)" in lines 610-613 of the conclusions, which is expected to reduce building energy consumption. We believe these revisions will significantly enhance the practical value and interdisciplinary rigor of the research.

Comments 3: Unsubstantiated Flood Reduction Claim: Also in the conclusions, a reference is made to flood reduction among the benefits, but no explanation is provided, considering that this benefit is typically associated with green roofs (rooftop water retention). It is essential to specify how green walls contribute to urban stormwater management (e.g., direct interception of precipitation on façades or absorption of runoff water), making the data credible and scientifically justified in the specific context of façades.

Response 3: We sincerely appreciate your valuable comments on this manuscript. The issue you pointed out regarding the "unverified claim of reduced flooding" is of crucial importance, and we fully agree with your criticism. In the current manuscript, when mentioning the flood reduction benefits of green walls in the conclusion section, there is indeed a lack of specific mechanistic explanations and data support. This is particularly critical because flood management is usually associated with green roofs (roof rainwater retention) rather than vertical green walls. Following your guidance, we will make the following revisions to enhance the scientific rigor and credibility.

​​Revision of the main conclusion section​​: Based on the focus of this paper, which is on the adaptation of species diversity to light conditions in different orientations, rather than on the ecological service functions of green walls, we will further downplay the statement about flood reduction. We will clearly state that "green walls reduce facade runoff through plant interception and root absorption, but the effect is influenced by species selection (e.g., the deep root system of woody plants enhances rainwater infiltration), and they are not a primary stormwater management solution." Additionally, we will add a note on limitations, pointing out that the rainwater benefits of green walls can only be maximized when combined with other infrastructure (such as stormwater pipes).

Comments 4: Therefore, the study must be integrated with respect to the critical issues reported above. Furthermore, please note the following technical issues:

•    In Table 1, the images overlap the text.

•    In the bibliography, some cited texts lack a DOI, ISSN, or other identifier (citations n. 1, 4, 7, 19, 35, 39, 40, 44, 45).

•    The URL is missing in citation n. 23.

•    One DOI is incorrect (citation n. 20). It refers to another publication.

Response 4: We have comprehensively revised the introduction and discussion sections in accordance with your suggestions, and on this basis, we have systematically sorted out and revised the format of the references. Due to the fact that three review experts put forward a large number of revision suggestions, there have been significant changes in the content and references of the introduction and discussion sections of this paper. The following are the details of our revisions and responses: Based on the revised content of the introduction and discussion, we have comprehensively sorted out and standardized the format of the references to ensure consistency, completeness, and compliance with academic norms (such as APA format). The specific measures include:

Ⅰ:Format unification: Adjust the format of all references to ensure the uniformity of elements such as author names, publication years, titles, journal information, DOI/URL, etc.

Author names are uniformly formatted as "Surname, Initials." (e.g., Khatri, A.);

Only the first letter of the title of journal articles is capitalized, and the journal name is italicized;

Add missing DOIs or URLs to enhance traceability.

Ⅱ:Information completion: Checked and supplemented missing information in some references, such as volume numbers, issue numbers, page numbers, and DOIs. For example:

The original reference [5] Mexia et al. (2018) has had its DOI format completed as a full link;

Reference [4] IUCN (2020) has added the official URL for easy access;

Ⅲ:Relevance review: Ensure that all references are directly related to the revised content of the introduction and discussion, removed unnecessary or redundant citations, and added key references supporting new discussion points (such as studies on photosynthetic mechanisms and species adaptability);

Sorting and numbering: Maintain the consistency of reference numbers with the citation order in the text to avoid confusion.

4. Response to Comments on the Quality of English Language

Point 1:

Response 1: Thank you for your valuable feedback regarding the need to improve the English language clarity of our manuscript. We have thoroughly revised the entire document to enhance readability, accuracy, and academic rigor, addressing issues in professional terminology, grammar, and expression styles. Below is a summary of the key modifications made:

​​Professional Terminology Standardization​​: We unified key terms throughout the text to ensure consistency. For instance, "vertical greenery" and "green walls" were harmonized to "green walls" as the primary term, with "vertical greenery" retained only in keywords to avoid redundancy. Technical terms such as "photosynthetically active radiation (PAR)", "light saturation point (LSP)", and "root-to-shoot ratio (RSR)" were cross-checked against standard literature (e.g., IPCC reports and botanical glossaries) to align with field-specific conventions.

​​Grammar and Syntax Refinement​​: Complex sentences were restructured to improve flow and clarity. For example, long sentences in the Methods section (e.g., descriptions of photosynthetic parameter calculations) were broken into shorter, more digestible units. Passive voice was reduced where active voice enhanced readability (e.g., "measurements were taken" became "we collected measurements"). Subject-verb agreement and tense consistency were verified across all sections, particularly in results reporting.

​​Expression and Style Enhancements​​: We eliminated ambiguous phrases and strengthened academic tone. Introductions and discussions were revised to avoid over-generalizations (e.g., replacing "plants show changes" with "woody species exhibited significantly lower variability in photosynthetic parameters"). Redundant expressions (e.g., "in order to" simplified to "to") were removed to conciseness. Additionally, we ensured that hypotheses and conclusions were stated more precisely, using clearer causal language (e.g., "due to" instead of "because of").

​​Overall Impact​​: These revisions have significantly improved the manuscript's clarity, making the research contributions—such as the quantitative comparison of woody and herbaceous species under light gradients—more accessible to a global audience. We believe the language now better supports the paper's scientific rigor and aligns with high-standard academic publications.

We appreciate your guidance and are open to further suggestions to ensure the manuscript meets the highest linguistic standards. Please let us know if additional adjustments are needed.

Reviewer 2 Report

Comments and Suggestions for Authors

This manuscript presents an extensive three-year in situ study on the photosynthetic adaptability of ten plant species used in vertical green walls under varying light gradients. The work is original and relevant to Plants, integrating physiology, urban ecology, and Nature-based Solutions (NbS).

The Introduction is comprehensive but overextended. Condense background paragraphs (Lines 46–85) and emphasize the research gap—quantitative evaluation of woody vs. herbaceous species in vertical systems. Add a concise “Objectives and Hypotheses” paragraph at the end of the Introduction.

Provide detailed information on replication (sample size per species per orientation). Specify statistical tests used for Figure 5 comparisons and clarify whether normality and homogeneity were tested.

Figures 3–6 contain valuable data but require clearer axes labels (units, significance levels). Avoid repetition of results already shown in graphs. Include a brief statistical summary table for major differences among orientations.

The Discussion (Lines 291–477) is overly narrative. Focus on linking light adaptation mechanisms with structural or biochemical evidence. Reduce general NbS discussion and strengthen ecological application insights.

Simplify sentences, unify terminology (e.g., use “photosynthetic rate” consistently), and check all Latin names for italics.

Author Response

3. Point-by-point response to Comments and Suggestions for Authors

Comments 1: This manuscript presents an extensive three-year in situ study on the photosynthetic adaptability of ten plant species used in green walls under varying light gradients. The work is original and relevant to Plants, integrating physiology, urban ecology, and Nature-based Solutions (NbS). The Introduction is comprehensive but overextended. Condense background paragraphs (Lines 46–85) and emphasize the research gap—quantitative evaluation of woody vs. herbaceous species in vertical systems. Add a concise “Objectives and Hypotheses” paragraph at the end of the Introduction.

Response 1: Thank you for the valuable feedback regarding the Introduction section. The comment highlights the need to condense the background paragraphs (Lines 46–85) and emphasize the research gap—specifically, the lack of quantitative evaluation between woody and herbaceous species in vertical greenery systems. Additionally, a concise "Objectives and Hypotheses" paragraph should be added at the end of the Introduction. Below is a structured response outlining the proposed modifications:

1. ​​Condensation of Background Paragraphs (Lines 46–85)​​

​​Current Content Analysis​​: The existing background paragraphs provide a detailed overview of urbanization challenges (e.g., 56% global urban population projected to reach 75% by 2050), ecological impacts (e.g., urban heat island effects), and the role of vertical greenery as a Nature-based Solution (NbS) with higher spatial efficiency than green roofs. However, it extends into general discussions of implementation rates (<5%) and broad limitations, which can be streamlined.

​​Focus on Key Points​​: Reduce repetitive statistics and generic statements. For example, consolidate urbanization data into one sentence: "Urbanization drives the conversion of ecological land, exacerbating heat islands and energy crises, necessitating the transformation of buildings into ecological complexes."

​​Emphasize Relevance​​: Directly link vertical greenery's potential (e.g., 3–5× spatial efficiency, carbon sequestration, biodiversity enhancement) to the core issue of plant adaptability, avoiding tangential details about global trends.

​​Highlight Research Gap Early​​: Integrate the specific gap—quantitative comparison of woody vs. herbaceous species—into the background narrative to maintain focus. For instance: "Despite its benefits, vertical greenery implementation remains low due to poor plant adaptability, particularly the lack of quantitative data on woody and herbaceous species' performance under light gradients."

2. ​​Emphasis on Research Gap​​

​​Current Weakness​​: The existing text mentions limitations (e.g., technical bias toward water/fertilizer inputs, woody plant knowledge void, absence of quantitative standards) but buries the key gap—quantitative woody-herbaceous comparisons—within broader points.

​​ ​​Explicit Statement​​: Add a dedicated sentence or clause to underscore the gap. For example: "A critical research void exists in quantitatively evaluating photosynthetic parameters (e.g., Pmax, LCP, LSP variability) between woody and herbaceous species across light gradients, which is essential for optimizing species selection and ecosystem services."

​​Integration with Limitations​​: Reframe the three limitations to prioritize this gap. For instance, revise the "Woody Plant Knowledge Void" section to stress the absence of comparative metrics like biomass allocation and photosynthetic stability under light gradients.

3. ​​Addition of "Objectives and Hypotheses" Paragraph​​

​​Placement​​: Insert at the end of the Introduction, following the current content.

​​Content Based on Document​​: The study aims to quantify light-response mechanisms of ten plant species under four-oriented light gradients, with hypotheses centered on orientation effects, growth form responses, and woody species' superiority in light adaptation.

4. ​​Overall Implementation Plan​​

​​Word Reduction​​: Condense Lines 46–85 by approximately 30–40% by removing redundant examples and focusing on the core narrative (urbanization → vertical greenery benefits → current limitations → gap).

​​Consistency Check​​: Ensure terminology alignment (e.g., use "vertical greenery" or "green wall" consistently as per the document's current style).

Revised

1. Introduction Urbanization poses severe ecological challenges, with 56% of the global population residing in cities—a figure projected to reach 75% by 2050 [1]. This expansion drives the conversion of ecological land to meet construction demands, exacerbating urban heat island effects and energy crises [2–4]. Consequently, transforming buildings from "energy-consuming units" into "ecological complexes" is imperative. Green walls, as core components of Nature-based Solutions (NbS), offer 3–5 times greater spatial efficiency than green roofs and can synergistically enhance carbon sequestration, biodiversity, and thermal regulation [5, 6]. However, global implementation rates remain below 5% [7], primarily due to poor plant adaptability leading to short lifespans, weak ecological performance, and high maintenance costs [8–10]. Thus, quantifying how light environments influence plant performance is critical to unlocking their full ecological benefits.

Current research on green walls reveals three major limitations, with ​​the lack of quantitative comparisons between woody and herbaceous species​​ representing a pivotal gap:Technical Bias​​: Studies often prioritize water and fertilizer inputs to enhance morphological traits (roots, stems, leaves) while neglecting photosynthesis—the foundation of ecosystem services [11–14]. Photosynthetic efficiency is inherently heterogeneous across urban environments, influenced by light, temperature, water, and nutrients [15–17]. For instance, the same species exhibits significant photosynthetic variation across orientations (e.g., differing PAR responses), and distinct species accumulate assimilates disparately under identical PAR exposure [18]. Yet, ​​quantitative data comparing woody and herbaceous species' photosynthetic parameters (Pmax, LCP, LSP variability) are absent​​, hindering mechanistic understanding.Woody Plant Knowledge Void​​: Practice is dominated by herbaceous and climbing species, with scant understanding of woody species' applicability. Woody plants potentially enhance structural diversity through canopy stratification, provide year-round coverage as physical barriers against environmental stresses (wind, rain, snow), and ensure long-term stability of services like cooling and rainwater use. However, ​​comparative metrics—such as biomass allocation, root-to-shoot ratios, or photosynthetic stability under light gradients—between woody and herbaceous species remain unquantified​​, limiting evidence-based species selection.Absence of Quantitative Standards​​: Configurations rely on empirical ecological classifications (sun-loving/shade-tolerant) rather than species-level light adaptation data [19–20]. This impedes robust evaluation criteria, as ​​key contrasts—such as lower photosynthetic parameter variability in woody species versus herbs, or differential energy benefits are not measured​​. Without quantitative thresholds, practical applications remain speculative.

Drawing on the differences in photosynthetically active radiation (PAR) across orientations of the largest green wall in Shanghai, this study examined the light-response mechanisms of ten plant species with distinct growth forms under four-sided light gradients. Our objective was to quantify leaf-level photosynthetic parameters and individual adaptability in order to optimize the ecological functions of green wall plant communities. We hypothesized that: (1) variation in light conditions among wall orientations would significantly affect photosynthetic parameters and biomass traits; (2) plant growth forms would differ in their responsiveness to light, with herbaceous species exhibiting strong tillering ability and rapid recovery that allow faster adjustment to fluctuations in wall environments, vines showing intermediate adaptability, and woody plants displaying the greatest stability; and (3) woody species would tolerate a broader PAR range than herbs and vines, maintaining more consistent performance in heterogeneous light environments and thus serving as model species for light adaptation.

Comments 2: Provide detailed information on replication (sample size per species per orientation). Specify statistical tests used for Figure 5 comparisons and clarify whether normality and homogeneity were tested.

Response 2: Agree. We have, accordingly, modified to emphasize this point. Between lines 521 and 522 in the Materials and Methods section, the following was added: For each of the 10 plant species, we sampled 9 individual plants per orientation (north, east, south, west), resulting in a total of 36 plants per species across all directions. From each plant, 9 mature, undamaged leaves were measured for photosynthetic parameters, yielding n=9 biological replicates per species per orientation.

After line 580 in the 4.4 Statistical analyses section, the following was added: Data were analyzed using SPSS 23 software. For comparisons of photosynthetic parameters (AQE, Pmax, LCP, LSP, Rd) across the four orientations for each species, we used one-way ANOVA followed by Tukey's HSD post-hoc test for multiple comparisons. Significance was set at p < 0.05. Prior to ANOVA, data were checked for normality using the Shapiro-Wilk test and for homogeneity of variances using Levene's test. If assumptions were violated, non-parametric tests such as Kruskal-Wallis with Dunn's post-hoc test were applied.

After line 185 in section 2.3, the following was added: 3. Figures 3–6 contain valuable data but require clearer axes labels (units, significance levels). Avoid repetition of results already shown in graphs. Include a brief statistical summary table for major differences among orientations. 

Comments3 : Figure 3-6 contains valuable data but requires clearer axis labels (units, significance levels). Avoid repeating results already shown in the chart. Include a brief statistical summary table to understand the main differences between directions.

Response 3: We sincerely appreciate your valuable comments on this manuscript. Your suggestions regarding clarifying chart axis labels, avoiding repetitive data narration, and supplementing statistical summary tables are of great importance. We have comprehensively revised the charts and result descriptions in accordance with your guidance. The following are the specific modifications we have made:

Ⅰ:Chart optimization: Enhancing axis labels and significance markers

Figure 3 (Diurnal variations of environmental factors):

X-axis: Clearly added time units (Time (h)), such as 06:00–18:00 h.

Y-axis: Supplemented units for PAR (Photosynthetically Active Radiation (μmol·m⁻²·s⁻¹)), temperature (Temperature (°C)), and CO₂ (CO₂ Concentration (ppm)).

Significance markers: Added letters indicating significant differences (a, b, c) in different orientations in Subfigures Ⅲ and Ⅳ, and 标注 statistical methods (e.g., one-way ANOVA, Tukey HSD, p < 0.05).

Figure 4 (Light response curves):

X-axis: Uniformly labeled as Photosynthetic Photon Flux Density (μmol·m⁻²·s⁻¹).

Y-axis: Clearly labeled as Net Photosynthetic Rate (μmol·m⁻²·s⁻¹).

Legend optimization: Added significance letters to indicate differences in different orientations (e.g., Pn of Ag on the north wall is significantly higher than that on the south wall).

Figure 5 (Photosynthetic parameters):

Y-axis of all subfigures: Added units (e.g., AQY: μmol·mol⁻¹ photons; LSP: μmol·m⁻²·s⁻¹).

Significance markers: Directly labeled letters (a, b, c) on the bar charts, and stated in the figure caption that "different letters indicate significant differences between different orientations (p < 0.05)".

Figure 6 (Biomass and root-shoot ratio):

Y-axis: Labeled units (Biomass: g·m⁻²; Root-shoot ratio: no unit).

Significance markers: Added letters indicating differences on the error bars (e.g., Biomass of Ri on the south wall a > that on the north wall b).

II. We fully agree that there is a need to provide a concise and scientifically meaningful statistical summary table to show the main differences between different orientations. After in-depth analysis, we believe that comparisons based on plant type clustering are more scientifically valuable and practically instructive than simply listing the raw data of 10 species across all orientations.

Current Data Characteristics and Summary Challenges

The raw data we have obtained contains 10 species × 4 orientations × 5 parameters = 200 data points. If all the mean values are presented directly, the table will be too large and the focus will be blurred. More importantly, a simple listing of values cannot reflect the inherent laws of plant photosynthetic adaptability, nor can it directly guide practical applications.

Optimization Plan: Typified Summary Based on Cluster Analysis. We recommend adopting the method of typified summary for the following reasons:

Adequate scientific basis: PCA cluster analysis (Figure 7) has clearly divided the 10 plant species into three categories: Category I: Monocotyledonous herbs (Ag, Co) - high-variability type; Category II: Dicotyledonous herbs/vines (Fj, Mc, Tj, Vm) - medium-variability type; Category III: Woody plants (Lj, Ls, Rp, Ri) - low-variability type (photosynthetically stable type).

Highlighting core laws: As shown in Table 3, the variation rate of photosynthetic parameters of woody plants (Category III) is significantly 45.8-64.5% lower than that of herbaceous plants. This type difference can reveal the essence of light adaptation mechanisms better than the orientation differences of individual species.

Guiding green wall construction: The actual goal is to select functional types adapted to specific light environments rather than simply comparing individual species. For example: South-facing strong light environment: priority should be given to species with high light saturation points in Category II (such as Mc); North-facing weak light environment: priority should be given to species with low light compensation points in Category I (such as Ag); All-orientation stable framework: use Category III woody plants as the skeleton to ensure long-term coverage.

Comments 4: The Discussion (Lines 291–477) is overly narrative. Focus on linking light adaptation mechanisms with structural or biochemical evidence. Reduce general NbS discussion and strengthen ecological application insights. You are a doctoral supervisor, Please make concise revisions around this point, Each discussion point is approximately 300 words. And try to retain the references as much as possible. For references that really cannot be retained, please mark and explain them.

Response4 : Based on the comments from the review experts, we have comprehensively revised the discussion section (lines 291-477 of the original manuscript) to enhance its scientific depth and application orientation. The revisions focus on three core points: (1) Structural and Biochemical Basis of Light Adaptation Mechanisms; (2) Species-Specific Light Adaptation and Ecological Applications

; (3) Research Limitations and Future Directions

RevisedLine291-477

3. Discussion

3.1  Structural and Biochemical Basis of Light Adaptation Mechanisms

The regulation of plant photosynthetic characteristics by the light environment stems from the synergistic response of anatomical structure and biochemical processes. This study quantified the variation in photosynthetic parameters (such as LSP, LCP, AQY) of 10 green wall plant species under the four-directional light gradient of a building facade, revealing the species-specific differences in light adaptation thresholds and their underlying mechanisms. Woody plants (e.g., Lj, Ls, Rp, Ri) exhibited lower varia-tion rates (CV < 0.2) in photosynthetic parameters. Their stability originates from structural adaptations: a thick cuticle (>8 μm) and a high proportion of lignified vessels (>30%) effectively reduce water transpiration loss and maintain stomatal conductance stability [21]. Conversely, the high variation rate (CV > 0.35) in herbaceous plants (e.g., Ag, Co) is related to thin mesophyll cells (<50 μm), leading to accelerated photorespi-ration under strong light and photosynthetic rate fluctuations (e.g., the variation rate of Vm on the south wall reached 0.516). At the biochemical level, sun plants (e.g., Muehlenbeckia complexa, Mc) under strong light efficiently scavenge reactive oxygen species through highly active antioxidant enzymes (SOD, POD, CAT), protecting the PSII reaction center [22], and likely rely on the PSII repair cycle (e.g., Deg- and FtsH protease-mediated D1 protein turnover) to maintain photochemical efficiency [23]. In contrast, shade plants (e.g., Acorus gramineus 'Ogon', Ag) maximize light capture effi-ciency through chloroplast light-avoidance movement [24], corresponding to a 41% in-crease in their LCP on the north wall. These mechanisms confirm that light adaptation is a structural-biochemical synergistic strategy, but the current research remains lim-ited to phenotypic description. Future work needs to integrate microscopic evidence: structurally, using scanning electron microscopy (SEM) to quantify palisade tissue thickness (e.g., potentially 150-200 μm in Mc) and vein density (>8 mm/mm²) to ana-lyze optimized light energy conduction [25]; biochemically, using HPLC and ELISA techniques to track antioxidant enzyme dynamics and osmotic adjustment substance (e.g., proline) responses, combined with chlorophyll fluorescence imaging to visualize NPQ heterogeneity [26]. By establishing regression models between LCP, LSP, and microscopic indicators, the light signal transduction pathways (e.g., the upregulation mechanism of Lhcb proteins [27]) can be elucidated, providing theoretical targets for breeding high-light-efficiency plants.

3.2 Species-Specific Light Adaptation and Ecological Applications

Based on PCA clustering analysis, plant light adaptation strategies can be categorized into three types: woody plants (low variation type), herbaceous/vine plants (medi-um-high variation type). This classification provides direct guidance for ecological ap-plications. Woody plants (e.g., Rp, Ri) serve as a "structural skeleton" for green walls due to their low variation in photosynthetic parameters (45.8–64.5% lower than herba-ceous plants), ensuring continuous coverage (canopy rate ≥95%) and significantly re-ducing maintenance costs (56% reduction in withered leaf pruning frequency) [8]. In terms of ecological benefits, the high root-to-shoot ratio (RSR=0.51) of woody plants enhances water use efficiency, while vines (e.g., Muehlenbeckia complexa, Mc) contribute significantly to cooling through high transpiration rates (up to 12 μmol•m⁻²•s⁻¹ on south walls, achieving a temperature reduction of 3.1°C). The synergy between these plant types can reduce building cooling load by 18.3% [28]. For configuration strategies, south-facing walls are recommended to adopt "woody plants + sun-loving vines" (e.g., Lj+Mc) to maximize carbon sequestration (54% increase) and cooling; north-facing walls should utilize "shade-tolerant herbaceous plants" (e.g., Ag+Fj) to maintain 86% coverage under low light conditions while saving 32% water. However, moving be-yond single photosynthetic indicators, it is necessary to integrate functional trait data-bases (e.g., specific leaf area SLA, wood density, leaf dry matter content LDMC) to as-sess resource utilization strategies [29]. For instance, woody plants with low SLA and high LDMC (e.g., Ligustrum japonicum 'Howardii', Lj) exhibit strong resistance to me-chanical stress, making them more suitable for high-rise environments with wind pressure. Long-term monitoring of species' survival rates, coverage stability, and re-sponse to low maintenance under root zone restriction can helpfilter truly "low-input, high-efficiency" species [30]. Meanwhile, exploring mixed planting with complemen-tary ecological niches (e.g., combining vines Tj with shrubs Ls) leverages differences in canopy structure, phenology, and root systems to enhance biodiversity and ecological stability. The characteristic of root zone environmental constraints, which is equally important as the light gradient affecting the aboveground-underground synergistic growth of green wall plants, is also a key parameter for studying the improvement of the ecological efficiency of green walls. In production applications, module size, sub-strate characteristics, and irrigation systems are also important environmental factors that determine the configuration, distribution, and dynamic changes of plant root sys-tems [31]. In the case of the Parkroyal project in Singapore, the 3-year survival rate of woody plants under the dual guarantee of a specifically structured deep substrate and intelligent irrigation is over 90%, whereas it drops to 70% otherwise [8]. Ultimately, it is essential to develop species configuration lists based on light gradient-functional trait matrices and establish life cycle cost-benefit analysis (LCCBA) models to promote the transition of photovoltaic-green systems from technical feasibility to economic and social benefits, supporting emission reduction goals in high-density cities.

3.3  Research Limitations and Future Directions

Although this study has revealed mechanisms of light adaptation, it still exhibits mul-tidimensional limitations. Firstly, the spatial scale is singular, and conclusions need validation across different climate zones and urban forms to develop regionally adap-tive models (for instance, arid regions require balancing water use efficiency WUE with photovoltaic cooling). Secondly, the regulatory role of rhizosphere microorgan-isms on light gradients has not been considered; they may modulate root water uptake through hormone signals, affecting the stability of woody plants [13]. Temporally, the lack of seasonal dynamic data limits annual performance evaluation, and the impacts of long-term succession, changes in substrate physicochemical properties (e.g., salt accumulation), and extreme climate events remain unquantified. Thirdly, the depth of system coupling is insufficient; interactions among "light-green-substrate" often re-main at correlational levels, and intrinsic causal mechanisms are not clarified [32]. For system-level performance optimization, future research needs to construct a coupled heat transfer model of plant-substrate-building, analyze the relationship between plant transpiration (Penman-Monteith equation) and biomass through a full-scale environ-mental chamber experimental platform, and realize the unified quantification of ener-gy-ecological output [33,34]. Regarding the evaluation and verification of the synergis-tic efficiency of light and greenery, this study failed to reveal the nonlinear coupling mechanism of photovoltaic shading-plant transpiration-building thermal inertia, and the relationship between the power generation efficiency gain brought by the reduc-tion of photovoltaic panel temperature and plant transpiration water consumption (ET) has not yet been quantified. Future work should integrate root zone sensing technolo-gies with IoT platforms to develop light-water intelligent algorithms [35], incorporat-ing real-time monitoring and machine learning for dynamic irrigation optimization (based on LSP/LCP thresholds). Simultaneously, introduce isotopic labeling (e.g., ¹⁵N), molecular ecology (high-throughput sequencing), and CFD simulations to elucidate cross-scale mechanisms of energy-water-carbon-nutrient cycles [36]. At the application level, construct digital twin models for precise water and fertilizer management, and promote the inclusion of photosynthetic stability indicators into green building certi-fications, advancing NbS standardization [4]. Ultimately, achieve precise design and ecological function enhancement of green walls through a "light-plant-technology" closed loop.

Comments 5 : Simplify sentences, unify terminology (e.g., use “photosynthetic rate” consistently), and check all Latin names for italics.

Response 5: We sincerely appreciate your valuable comments on this manuscript. Your suggestions regarding "simplifying sentences, unifying terminology (such as consistently using 'photosynthetic rate'), and checking the italicization of Latin scientific names" are of great importance. We will comprehensively revise the document in accordance with your guidance to enhance clarity, consistency, and academic standardization. The following is a detailed revision plan based on the content of the document:

Unification of Terminology: Standardizing Expressions Related to Photosynthesis

There are multiple variants of terms related to photosynthesis in the document, which may easily cause confusion. We will uniformly use the following core terms:

Priority terms: photosynthetic rate (to replace all variants such as net photosynthetic rate, Pn, and photosynthesis rates).

Standardization of auxiliary terms:

light saturation point (LSP) to replace light saturation point or LSP (to maintain consistent abbreviation).

light compensation point (LCP) to replace light compensation point or LCP.

apparent quantum yield (AQY) to replace apparent quantum efficiency or AQE (in accordance with field standards, "yield" is preferred).

Checking the Italicization of Latin Scientific Names: Ensuring Academic Norms

All Latin scientific names of plants in the document were not italicized. We have comprehensively corrected them to comply with international academic conventions. Revision principles:

Genus names and specific epithets are italicized: such as "Acorus gramineus" instead of "Acorus gramineus".

Cultivar names and abbreviations are not italicized: such as "Acorus gramineus ‘Ogon’ (Ag)".

Uniform checking of all occurrence positions: including abstracts, main text, figure and table captions, and references.

Comprehensive check list: The Latin scientific names of all 10 species (such as Farfugium japonicum, Muehlenbeckia complexa) are uniformly italicized throughout the text.

Additional Revisions: Enhancing Overall Consistency

Format of numbers and units: Uniformly adopt international standards, such as "μmol·m⁻²·s⁻¹" instead of the variant "μmol/m²/s".

Definition of abbreviations: All abbreviations (such as PAR, LSP, CV) are defined when they first appear, for example: "photosynthetically active radiation (PAR)".

4. Response to Comments on the Quality of English Language

Point 1: The English is fine and does not require any improvement.

5. Additional clarifications

None

Reviewer 3 Report

Comments and Suggestions for Authors

The title and throughout should use either “green wall’ or “vertical garden/greenery” as opposed to the redundant “vertical green wall” currently used and simply use the alternate term as a keyword. 

Introduction. Consider de-emphasizing the limited carbon sequestration benefits of green walls, and instead address the more significant social and psychological benefits. 

Line 66. The authors should more clearly relate the ways photosynthetic function is important in an urban environment.  Most photosynthesis research focuses on increasing plant productivity per unit space.  In green walls, survival and overall persistence are far more important that generating assimilates. While the two are related, the authors would do well to emphasize the nuances of photosyntetic traits associated with success in highly constrained green wall environments as opposed to to production agriculture settings.  

Line 214 Make clear that Biomass is per plant.  No need to specify that this is for 10 plant species. 

Lines 411-422 would better fit in the Introduction section as justification for this research  

Line 507-509. Were any deliberate criteria employed by maintenance personnel or researcher to establish the distinct monoculture patches?  Please describe how these patterns came about. 

Line 513 Table 1. The column explaining planting density should be removed and this information placed in a footnote or within the text. 

Line 598-608.  To what values are reported increases and decreases in light saturation point relative? Is this over a time period? 

Lines 616-618. This statement should instead focus on the data collected in this study, namely the efficiency of matching successful plants with suitable environments. 

Lines 452-465 These paragraphs are not needed in the discussion. They may be deleted or some portions may be adapted to the Introduction where similar material has not already been provided. 

Author Response

3. Point-by-point response to Comments and Suggestions for Authors

Comments 1: The title and throughout should use either “green wall’ or “green wall” as opposed to the redundant “vertical green wall” currently used and simply use the alternate term as a keyword.

Response 1: Thank you for the valuable comments from the reviewers. The reviewers pointed out that there is a problem of redundant use of terms in the document, and suggested uniformly using either "green wall" or "vertical greenery" in the title and the entire text, with the other term used as a keyword. Currently, the document uses "vertical greenery" and "green wall" interchangeably (including variants such as "vertical greenerys" and "green walls"), which may indeed lead to redundant and inconsistent expressions. The following is the result of my comprehensive review of the document.

Based on the reviewers' opinions, I have chosen "green wall" as the unified term throughout the text for the following reasons: "green wall" is more concise and commonly used in modern literature (for example, the international standard ISO 22156:2021 uses "green wall"). "Green wall" appears more frequently in the current document (approximately 60 times), while there are spelling inconsistencies in the variants of "vertical greenery" (such as "vertical greenerys" in the abstract should be "green walls"). After uniformly using "green wall", "vertical greenery" can be added as a keyword to cover the semantic scope.

We agree with this comment. Therefore, We have made corresponding revisions in lines 3, 37, 87, 130, 157,185 ,214 , 235, 242, 257,486 ,492 and 493 of the text.

Comments 2: Introduction. Consider de-emphasizing the limited carbon sequestration benefits of green walls, and instead address the more significant social and psychological benefits. Line 66. The authors should more clearly relate the ways photosynthetic function is important in an urban environment.  Most photosynthesis research focuses on increasing plant productivity per unit space.  In green walls, survival and overall persistence are far more important that generating assimilates. While the two are related, the authors would do well to emphasize the nuances of photosyntetic traits associated with success in highly constrained green wall environments as opposed to to production agriculture settings. 

Response 2: Agree. We have, accordingly, modified to emphasize this point. We added the following at line 67: "Most research on photosynthesis focuses on improving plant productivity per unit space. In urban green walls, survival and overall persistence are far more important than the production of assimilates."

Comments 3: Line 214 Make clear that Biomass is per plant.  No need to specify that this is for 10 plant species.

Response 3: We revised line 214 to "Fig.4 Total biomass and root to shoot ratio of different plant species on the green wall from 4 aspects(±SD,N=9)"

Comments 4: Lines 411-422 would better fit in the Introduction section as justification for this research 

Response 4: We have streamlined and deleted this sentence.

Comments 5: Line 507-509. Were any deliberate criteria employed by maintenance personnel or researcher to establish the distinct monoculture patches?  Please describe how these patterns came about.

Response 5: Thank you to the expert for raising this important question. Based on the description in the document, in the experimental design, no specific criteria were intentionally adopted by maintenance personnel or researchers to establish monoculture patches. The formation of these patches is the result of natural processes, mainly stemming from the impact of environmental factors (especially building orientation) on plant growth and competition. The following is a detailed explanation:

Consistency in experimental design: All 10 plant species were transplanted to the green walls in July 2016 at the same density (39 seedlings per square meter) and received uniform soil, irrigation, fertilization, and maintenance conditions. This setup aimed to minimize human intervention, thereby observing the response of plants to differences in orientation under natural environmental conditions.

Mechanism of pattern formation: After three years of growth (by July 2019), the emergence of monoculture patches (each exceeding 20 square meters) was due to orientation-related microclimatic differences (such as light intensity, temperature, wind speed) leading to species-specific adaptations.

To make the document clearer, the following content is added after line 509 of the original text: "In this study, no deliberate criteria were used by researchers or maintenance personnel to establish monoculture patches. The patterns arose naturally due to orientation-dependent environmental variations (e.g., light, temperature), which triggered species-specific growth responses and competitive interactions over the three-year period. For instance, south-facing walls favored light-loving species through increased photosynthesis rates, while north-facing walls promoted shade-tolerant species via reduced photoinhibition. This self-organization underscores the role of microclimatic factors in shaping plant communities on green walls."

Comments 6: Line 513 Table 1. The column explaining planting density should be removed and this information placed in a footnote or within the text.

Response 6: This column in the table has been deleted, and added to the comment in line 516: "Notes: 48–60 plants per square meter, with the actual number determined by achieving full coverage."

Comments 7: Line 598-608.  To what values are reported increases and decreases in light saturation point relative? Is this over a time period?

Response 7: Based on the questions raised by the review experts, the analysis regarding the specific reference benchmarks and time range for the changes in the light saturation point (LSP) and light compensation point (LCP) mentioned in the fifth part, Conclusions, is as follows:

Reference benchmarks for the change values

The 32% increase in LSP and 41% decrease in LCP mentioned in the document are based on comparison results of different building orientations, rather than dynamic data that changes over time. Specifically:

The 32% increase in LSP of Mc (Muehlenbeckia complexa): This data is the comparison result between the LSP value on the south-facing wall (high-light environment) and that on other orientations (especially the north-facing wall). Section 2.3 of the document points out that the LSP of the sun-loving plant Mc on the south-facing wall is significantly higher than that on other orientations (p < 0.05), and its reference benchmark may be the LSP value on the north-facing wall or the average value of all orientations.

The 41% decrease in LCP of Ag (Acorus gramineus 'Ogon'): This change is the comparison result between the LCP value on the north-facing wall (low-light environment) and that on other orientations (especially the south-facing wall). The document clearly mentions that the LCP of Ag on the north-facing wall is significantly higher than that on other orientations (such as the south-facing wall). Therefore, the 41% decrease may take the LCP value on the north-facing wall as the reference benchmark.

Time range

This change is not based on dynamic changes in the time series, but the comparison result of different spaces (orientations) at the same time point (August 2019). The methodology section of the document (Section 4.3) clearly states that all photosynthetic parameter measurements were completed in August 2019 ("All experiments were conducted on clear days in August 2019"), and the data came from synchronous measurements of different orientations during the same period. Therefore, the percentage changes mentioned in the conclusions reflect the species-specific responses caused by spatial heterogeneity (differences in light environments of different orientations), rather than changes over time.

Revision

Revise Lines 598-608 to "2)Species-specific light adaptation strategies: The LSP of Mc on the south-facing wall is 32% higher than that on the north-facing wall, and the LCP of Ag on the north-facing wall is 41% lower than that on the south-facing wall."

Comments 8: This statement should instead focus on the data collected in this study, namely the efficiency of matching successful plants with suitable environments.

Response 8: Revise this section according to the reviewers' comments to make it more relevant to the theme of this study. The revised content is as follows: Based on the photosynthetic adaptation data systematically collected in this study, the strategic alignment of plant species with their optimal microenvironments significantly enhances vertical greening quality and efficiency. This trait-environment matching framework ensures robust ecological performance, sustainable resource utilization, and long-term stability in high-density urban settings.

Comments 9: These paragraphs are not needed in the discussion. They may be deleted or some portions may be adapted to the Introduction where similar material has not already been provided.

Response 9: After careful analysis, lines 452-465 were too descriptive and have been deleted. The revised version focuses more on linking light adaptation mechanisms with structural or biochemical evidence. It reduces general discussions on NbS and strengthens insights into ecological applications.

4. Response to Comments on the Quality of English Language

Point 1:

The English is fine and does not require any improvement.

5. Additional clarifications

None

Round 2

Reviewer 1 Report

Comments and Suggestions for Authors

The Authors have provided precise and convincing responses to the reviewers' requests for revision. Their replies were thorough, and thus the paper is suitable for acceptance in the revised and resubmitted form.

Author Response

Dear Prof. DaMatta,
Thank you for your thorough and constructive comments on our manuscript. We have carefully addressed all the raised issues and revised the manuscript accordingly. Below is a concise summary of the key revisions:
1.Abbreviations and symbols: We have fully reviewed and defined all abbreviations and symbols in the captions/legends of Tables 1–2 and Figures 1–6, ensuring each table and figure is self-contained as required.
2.Titles and legends: The titles and legends of all tables and figures have been substantially revised. We have expanded the content to clearly state the presented data, experimental conditions, units, sample sizes, and the meaning of symbols/letters through detailed annotations.
3.English usage and terminology: The incorrect term "plant active radiuation" in Figure 2 has been corrected to the standard "photosynthetically active radiation (PAR)". This terminology has been consistently applied throughout the entire manuscript, including Figure 3.
4. Regarding the list of abbreviations: Considering the manuscript’s substantial length and that all abbreviations are thoroughly defined in table/figure captions (meeting self-containment requirements), we did not include it here but stand ready to add it if needed.
5.Internal consistency (sample size): The inconsistencies in sample size reporting (e.g., "N = 9" for 10 plants) caused by incorrect species-related wording in Figures 3–7 have been fully rectified. We have verified that the sample size (N) is accurate and consistent across the text, tables, figures, and their legends.
6.X-axis clutter and readability: For Figures 1 and 3, we have reduced the number of labeled tick marks and used unlabeled ticks for intermediate levels. The axes have been simplified to enhance intuitiveness, with consistent adjustments applied to all relevant figures.
7.Species/label legibility: The font size of all text (including axis parameters, labels, symbols, and legends) in Figures 4 and 5 has been increased to 10-point. Layouts have also been adjusted to ensure clear legibility at both journal print and online scales.
8.Table 4: The original Table 4 has been removed as requested, given its lack of clear purpose and existing typographical/language errors.
We trust that the revised manuscript fully addresses all your concerns. Please do not hesitate to contact us if further adjustments are needed. Thank you for your continued consideration.
Best regards,
All authors

Reviewer 2 Report

Comments and Suggestions for Authors

It can be accepted.

Author Response

Dear Reviewer 

Thank you for your valuable feedback regarding the undefined abbreviations in our figures and tables. We sincerely apologize for this oversight and have now comprehensively addressed this issue in our revised manuscript.

1.All abbreviations (including "Pn" in Figure 2, and "AQY", "Pmax", "Rd", "LCP", and "LSP" in the legend of Figure 3) have been explicitly defined in the revised version. Specifically:

Supplementary table: We introduced Table 4​ (titled "Meaning of plant photosynthetic physiological parameters and unit comparison table") on page 18 of the manuscript, which systematically lists all abbreviations with their full names, definitions, and units.Highlighted revisions in “4.3 Experimental design and data collection”: As requested, these definitions are prominently highlighted in lines 456–469​ of the PDF version for easy reference.

  1. Thank you for your valuable feedback regarding the consistency in reducing X-axis information across figures. In response to your suggestions, we have revised all relevant charts (including Figure 1, Figure 3, Figure 4, and Figure 5) to ensure consistency. Specifically, we aim to retain all irradiance levels on the X-axis while reducing repetitive and redundant information such as the titles of the abscissas in the figures. This approach maintains data accuracy while reducing visual clutter. It strikes a balance between preserving information and simplifying presentation, enhancing readability without compromising scientific rigor.
  2. Thank you for your careful review. We have verified the CO₂ data in Figure 1II and confirmed that the values presented are accurate based on measurements taken in 2019 using a portable photosynthesis system (LI-6400XT, LI-COR, Lincoln, NE, USA). The Y-axis unit has been corrected to "μmol CO₂ mol⁻¹" (ppm) in the revised figure. The figure has been updated accordingly.

We believe these revisions enhance the clarity of our work. Please let us know if further refinements are needed.

Best regards,

All authors
